# Does Localization Inform Editing? Surprising Differences in Causality-Based Localization vs. Knowledge Editing in Language Models

Peter Hase[1,2]    Mohit Bansal[2]    Been Kim[1]    Asma Ghandeharioun[1]
[1]Google Research    [2]UNC Chapel Hill
{peter, mbansal}@cs.unc.edu
{beenkim, aghandeharioun}@google.com

## Abstract

Language models learn a great quantity of factual information during pretraining, and recent work localizes this information to specific model weights like mid-layer MLP weights [21]. In this paper, we find that we can change how a fact is stored in a model by editing weights that are in a different location than where existing methods suggest that the fact is stored. This is surprising because we would expect that localizing facts to specific model parameters would tell us *where* to manipulate knowledge in models, and this assumption has motivated past work on model editing methods. Specifically, we show that localization conclusions from representation denoising (also known as Causal Tracing) do not provide any insight into which model MLP layer would be best to edit in order to override an existing stored fact with a new one. This finding raises questions about how past work relies on Causal Tracing to select which model layers to edit [21, 22]. Next, we consider several variants of the editing problem, including erasing and amplifying facts. For one of our editing problems, editing performance does relate to localization results from representation denoising, but we find that which layer we edit is a far better predictor of performance. Our results suggest, counterintuitively, that better mechanistic understanding of how pretrained language models work may not always translate to insights about how to best change their behavior.[1]

## 1  Introduction

Language models learn a variety of facts about the world during pretraining that can be elicited via natural language prompts [28]. Recent work explores how these facts are stored in model weights and expressed in response to particular prompts, suggesting that MLP weights act as key-value memories that support factual association [12, 21, 13]. Besides improving our scientific understanding of pretrained language models, this kind of investigative work may enable the design of better model editing methods for injecting new facts into model weights, and indeed it has been used to motivate the ROME and MEMIT model-editing methods [21, 22]. These recent methods set a new state of the art for weight edits that successfully rewrite stored facts in language models. Model editing methods could be broadly useful for correcting factual errors in pretrained models, avoiding morally undesirable outputs, and updating models with changing knowledge over time.

The connection between *localization* (identifying components of a model responsible for a certain behavior) and *editing* (changing model components in order to change model behavior) is predicated on the reasonable assumption that one should go about editing a model by first localizing a behavior to a specific component and then choosing to edit that particular component. In the case of ROME and

---

[1]Code for all experiments is available at https://github.com/google/belief-localization

37th Conference on Neural Information Processing Systems (NeurIPS 2023).

MEMIT, localization is done via Causal Tracing, which measures the information content of hidden representations, and editing is done by treating MLP weights as linear associative memories and injecting new key-value memories into the weights. Meng et al. [21, 22] choose to edit early MLP layer(s) based on results from Causal Tracing showing the largest causal effects on average in early layers.

**Surprisingly, the assumption that one should change the knowledge in a model by editing the weights where it is stored turns out to be false.** In fact, localization results from Causal Tracing are statistically uncorrelated with the success of an edit injecting a new fact into MLP weights. Using the CounterFact dataset from Meng et al. [21] with a GPT-J model [35], we show that (1) not only is a substantial fraction of factual knowledge stored outside of the range of layers edited by ROME/MEMIT (see Fig. 1), (2) the correlation between Causal Tracing results and edit success is near zero (for several editing methods including ROME, MEMIT, and Adam-based finetuning). We note that this is surprising largely because ROME and MEMIT *do* work well for editing facts, in spite of Causal Tracing often suggesting knowledge is stored elsewhere than early-to-mid-layer MLP weights.

In the face of this result, we attempt to recover the connection between tracing-based localization and editing by introducing four variants of the default model editing problem. Each variant differs in terms of the input, target, or objective used in the editing problem. One variant we introduce, called Fact Forcing, is designed to match Causal Tracing along these three factors. Specifically, Fact Forcing uses a noised input and involves maximizing the probability of the correct target output, just like Causal Tracing. We find that tracing results *are* related to edit success for Fact Forcing. However, even for this variant, it is still better to ignore the tracing results and always choose an early-to-mid-layer MLP weight for editing. We conclude that, although Causal Tracing is a reasonable localization method that has yielded insight into how models store factual information, this insight does not actually indicate which model layers we should edit in order to manipulate what facts are stored in language models.

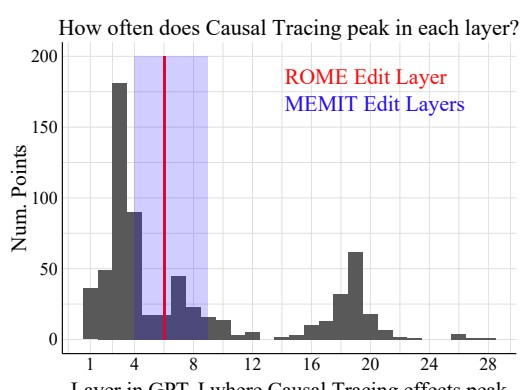

Figure 1: We visualize where 652 facts known by GPT-J are stored within the model, as localized by Causal Tracing. Model editing methods like ROME and MEMIT can successfully change knowledge in LMs by editing layers 4-9. But many facts appear to be stored outside of this range, e.g. at layers 1-3 and 16-20. What about these facts?

To summarize, our conclusions are as follows:

1. We find that model edit success is essentially unrelated to where factual information is stored in models, as measured by Causal Tracing. Robustness experiments generalize this result across causal localization methods, editing methods, editing metrics, models, and datasets.

2. To reconnect localization with editing performance, we introduce four variants of a standard model editing problem, including Tracing Reversal, Fact Erasure, Fact Amplification, and Fact Forcing.

3. Edit success and tracing effects correlate best in the Fact Forcing setting. However, tracing effects explain only a small fraction of the variance in editing performance, while the choice of edit layer is a much more important factor. This suggests that, surprisingly, localization insights from Causal Tracing are not useful for choosing which model layer to edit.

## 2   Related Work

**Localization.** A long line of work aims to interpret what certain hidden representations represent, or, in the reverse direction, to understand how a given concept is represented in a model. Both of these efforts aim to localize behaviors to specific model components. We group these methods based on the kinds of model components they consider (e.g. layers, neurons, etc.).

Many works focus on individual layers or weight matrices [37, 31, 9, 33, 11]. In this paper, we adopt the layer-wise localization method from Meng et al. [21] known as Causal Tracing, which estimates the information content of a set of representations via a denoising operation. We specifically focus on MLP layers given evidence of their role in factual association [12, 21, 13].

Related to analysis at the layer level, other work aims to localize concepts to directions in a latent space, dating back to work interpreting directions in word vector space [23, 17, 41, 14, 38, 5]. One might also place "key-value memory" theories of weight matrices in this category since a key vector represents a direction in the latent space [1, 32, 12, 21].

Neurons, meanwhile, are the most common focus of localization analysis. Past work explores the functions of groups of neurons and subnetworks [27, 6, 10, 4] or simply individual neurons [29, 40, 18, 2, 34, 26, 8, 19, 3, 16, 7, 36].

**Relating Localization to Editing.** Many works on localization validate the quality of their conclusions by editing neuron activations or layer weights corresponding to a particular concept, then checking that the network behavior changes appropriately. For example, Dai et al. [8] check that their "knowledge neurons" have localized a specific fact by amplifying or suppressing the expression of that fact via adjusting the corresponding neuron activations. Altogether, we find many localization analyses are validated by editing models in suggested locations [29, 18, 2, 26, 34, 8, 19, 7, 36, 4] or directions in the latent space [23, 1, 32, 21].

Changing model behavior by editing components suggested by localization seems like a reasonable validation step. However, in isolation, it paints an incomplete picture that has led to misleading interpretations about the connections between localization and editing. Such experiments alone do not show whether editing *that specific component* is (1) successful in proportion to the strength of the localization, (2) necessary to achieve the desired behavior, or (3) the best option for editing. In particular, these experiments do not show whether the same change in behavior can be achieved *elsewhere in the network*. Meng et al. [21] consider this question by measuring editing success across layers, averaged across data, then comparing the results with Causal Tracing conclusions also averaged across data. However, as we show, more fine-grained analysis at the datapoint level reveals the unexpected result that tracing results are unrelated to edit success. We are not aware of any work that primarily investigates the connection between localization and editing or that demonstrates better model editing at locations elsewhere in the network than those suggested by localization analysis.

# 3 Notation and Background

## 3.1 Data Notation

Following Meng et al. [21], we consider facts of the form $(s, r, o)$, where $s$ represents a subject entity (e.g. *Paris*), $r$ a binary relation (e.g. *is located in*), and $o$ an object (e.g. *France*) for which the tuple $(s, r, o)$ represents a factual assertion about the world. In the CounterFact dataset [21], each datapoint is a prompt $P$ for some fact $(s, r, o)$. So, $P$ might be "Paris is located in" or "Paris is situated in," to be completed by the object $o$ to form a true statement. In an abuse of notation, we will often use $s$ and $r$ to refer to textual representations of a subject and relation, for instance by writing a model's conditional probability as $p_\theta(\cdot | s, r)$ instead of $p_\theta(\cdot | P)$. We do so in order to more easily indicate when an input is provided where the subject or relation has been manipulated (described next).

We make use of a few variations of the data for the fact $(s, r, o)$. The additional variables include:

1. $s^*$ is a "neighboring" entity to the subject $s$ (similar to $s$) for which $(s^*, r, o)$ is a true fact like $(s, r, o)$. In CounterFact, "Marseille" is a neighboring entity to "Paris."
2. $r^*$ is a paraphrase of the relation $r$, such as "is situated in" for "is located in."
3. $s_{noise}$ is a noised representation of the subject $s$. We add Gaussian noise to the token embeddings of $s$, following Meng et al. [21].
4. $o_{false}$ is an object that incorrectly completes the tuple $(s, r, \cdot)$. CounterFact contains an $o_{false}$ for each datapoint, intended to be the new model output when evaluating model editing methods.
5. $o_{true}$, for clarity, is the object that correctly completes the fact $(s, r, \cdot)$, from CounterFact.

## 3.2 Causal Tracing

We give a brief description of Causal Tracing here and refer readers to Meng et al. [21] for more information (see Fig. 2 for an example visualization). Causal Tracing is a method for localizing information in the forward pass of an autoregressive Transformer to specific hidden representations. For a model with $L$ layers, the input is a prompt containing $T$ tokens (including a subject $s$ and

relation $r$). Given this input, the forward pass produces $T \times L$ layer outputs (one representation per $T$ tokens and $L$ layers). The algorithm aims to estimate the amount of information about the fact $(s, r, o_{true})$ that is contained in each of these representations. We denote the representation at token $t$ and layer $\ell$ as $v_{(t,\ell)}$.

The amount of factual information in $v_{(t,\ell)}$ is estimated by copying this representation into a different forward pass obtained from using a noised subject in the input:

$$\text{Tracing Effect} = p_\theta(o_{true}|s_{noise}, r, v_{(t,\ell)}) - p_\theta(o_{true}|s_{noise}, r)$$

where $s_{noise}$ indicates that we add Gaussian noise with $\sigma = 0.094$ to the token embeddings of $s$ following Meng et al. [21], and $v_{(t,\ell)}$ is the representation at token $t$ and layer $\ell$ in the forward pass on the original prompt $P = (s, r)$. The probability $p_\theta(o_{true}|s_{noise}, r, v_{(t,\ell)})$ is computed by (1) running the model forward pass on the noised prompt $P^* = (s_{noise}, r)$ until layer $\ell$, (2) *overwriting* the existing representation at token $t$ and layer $\ell$ with the representation $v_{(t,\ell)}$, then (3) computing the remaining $L - \ell$ layers as normal using this adjusted set of $T$ representations as input (adjusted at token index $t$). Thus, Causal Tracing estimates the information content of a representation in terms of its effect on the probability of the true target. The results from Causal Tracing show where the representations containing information about the true target are in the model forward pass.

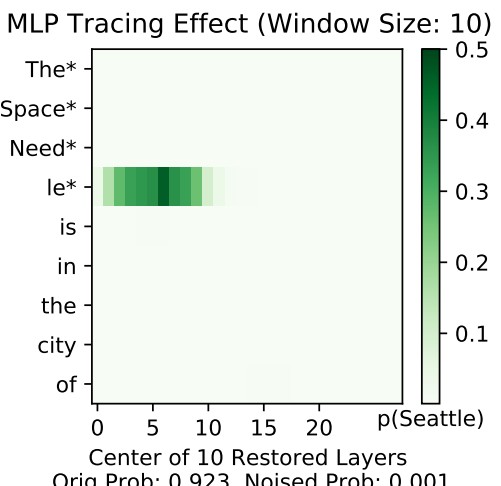

Figure 2: Visualizing Causal Tracing results over MLP layers with window size 10. Tokens with an asterisk are the noised subject tokens. Here, $p_\theta(o_{true}|s, r) = .923$ and $p_\theta(o_{true}|s_{noise}, r) = .001$.

**In practice, a *set* of representations from multiple adjacent layers is copied from the clean forward pass rather than a single layer's representation** (for instance, ten layers in Fig. 2). The size of this set is referred to as the *tracing window size*. A window size of, e.g., three implies that the tracing effect at layer $\ell$ estimates the amount of information contained in the three representations $v_{(t,\ell-1)}$, $v_{(t,\ell)}$, and $v_{(t,\ell+1)}$. See Appendix Figs. 10 and 11 for analysis of the parameter's effect. In this paper, we use a tracing window size of 5 by default, and we apply Causal Tracing exclusively to MLP layers, given evidence of their role in factual association [12, 21].

## 3.3 Model Editing with ROME

We describe the ROME editing method here since we use it in our analysis in Sec. 4, and later in Sec. 5 we outline additional editing methods we consider. For mathematical detail, see Meng et al. [21].

The input to ROME includes a prompt $P = (s, r)$ and a new desired output, which is always a false target $o_{false}$ in the CounterFact dataset. To change the model prediction to $o_{false}$, ROME applies a rank one edit to the down-projection matrix in a prespecified MLP layer in the model. The default layer in GPT-J

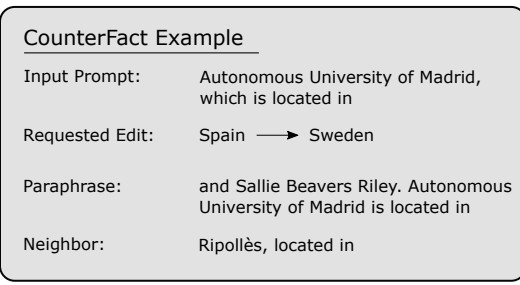

Figure 3: An example CounterFact datapoint.

is layer 6, following from averaged Causal Tracing results. ROME also makes use of covariance statistics of different subject representations obtained from a larger corpus as it edits individual facts. Overall, the method is designed to optimize the quantity $p_\theta(o_{false}|s, r)$ while aiming to satisfy some other constraints reflecting what a desirable model edit is (described in Sec. 3.4 next).

### 3.4 Editing Metrics

Editing methods are typically evaluated according to their ability to (1) change the model prediction on the input $P$ provided at runtime, (2) generalize appropriately to paraphrases of the prompt $P$, and (3) avoid over-generalizing to unrelated data [42, 9, 24, 15, 25]. We adopt metrics for each desideratum that we compute with available CounterFact data. Instead of the exact "magnitude" metrics from Meng et al. [21], we use normalized versions of each metric that we design to scale from 0 to 1 depending on whether the edit was maximally (un)successful, for purposes of making scores more comparable across data points. We denote the new edited weights of the LM as $\theta^*$ and its pre-edit weights as $\theta$. See Fig. 3 for an example of the kinds of data these metrics are computed on.

1. *Rewrite Score*. The rewrite score measures how much an edit improves the target probability $p(o_{false}|s, r)$ as a fraction of the maximum possible improvement:

$$\frac{p_{\theta^*}(o_{false}|s, r) - p_\theta(o_{false}|s, r)}{1 - p_\theta(o_{false}|s, r)}$$

2. *Paraphrase Score*. The paraphrase score measures the target probability using syntactical paraphrases as inputs, always preserving the exact subject wording:

$$\frac{p_{\theta^*}(o_{false}|s, r^*) - p_\theta(o_{false}|s, r^*)}{1 - p_\theta(o_{false}|s, r^*)}$$

   which is averaged over multiple available paraphrases per input $P$. The score measures whether edits properly generalize across semantically equivalent prompts.

3. *Neighborhood Score*. The neighborhood score measures whether edits change predictions for prompts with a similar subject $s^*$, the same relation $r$, and the same (true) objects. We scale the difference in probabilities so that 1 means the probability did not change (good), and 0 means it changed to the maximum extent possible (bad):

$$1 - \frac{|p_{\theta^*}(o_{false}|s^*, r) - p_\theta(o_{false}|s^*, r)|}{.5 + |p_\theta(o_{false}|s^*, r) - .5|}$$

   The score measures whether edits avoid *over*-generalizing from the prompt $P$ to different subjects.

## 4 Does Edit Success Follow From Localization?

Ostensibly, localization results should inform editing methods because it should help to know where information is stored in a model if you are going to manipulate the model's expression of that information. More specifically, if you wanted to inject a false belief $(s, r, o_{false})$ into a model (as defined in the ROME editing problem), it seems helpful to know which weights store the true fact $(s, r, o_{true})$, so that you could replace some stored representation of $o_{true}$ with that of $o_{false}$. This underlying assumption about editing models appears in much past work on localization, where editing is used to verify localization analysis (see Sec. 2). In this section, we investigate the validity of this assumption as it applies to autoregressive Transformers.

### 4.1 Experiment Design

The goal of our experiments is to determine, for a given datapoint, whether edit success *at a specific layer* aligns with the results from Causal Tracing at that layer (see Causal Tracing description in Sec. 3.2). We operationalize this outcome and explanatory variable as follows:

1. *Edit Success*. We primarily consider Rewrite Score as our measure of edit success, given that this is the main optimization objective of ROME. Note ROME achieves an average rewrite score of 99% at layer 6 of GPT-J and above 96% at layers besides the last layer of the model.

2. *Tracing Effect at layer $\ell$*. Since the output of Causal Tracing is a $T \times L$ grid of estimates, we obtain a single tracing effect per layer by taking the max across the $T$ token effects at each layer (i.e., we collapse the grid in Fig. 2 down to a single curve across layers). Like our other metrics,

we use a *fractional* tracing effect where 0 means the intervention had no effect and 1 means it fully restored the original probability $p_\theta(o_{true}|s, r)$:

$$\frac{p_\theta(o_{true}|s_{noise}, r, v_{(t,\ell)}) - p_\theta(o_{true}|s_{noise}, r)}{p_\theta(o_{true}|s, r) - p_\theta(o_{true}|s_{noise}, r)}$$

Lastly, note we use a tracing window size of 5 (smaller than the value of 10 used in Fig. 2).

## 4.2 Model and Data

We conduct our analysis with GPT-J [35] using the CounterFact dataset, similar to Meng et al. [21]. GPT-J is a 6 billion parameter autoregressive language model. We record editing performance at layers in {1, 5, 9, 13, 17, 21, 25, 28} as well as layer 6 (the default for ROME). Note ROME achieves an average rewrite score of 99% at layer 6 and above 96% at layers besides layer 28.

The CounterFact dataset includes datapoints consisting of a prompt, paraphrases, and neighboring points. For each point, a new (false) target is supplied for editing purposes. We show an example datapoint in Fig. 3. Note paraphrases intentionally include unrelated text preceding a syntactical paraphrase of the input, with the idea that this text should not affect the output. We select data for experiments from 10% of CounterFact, additionally filtering to a

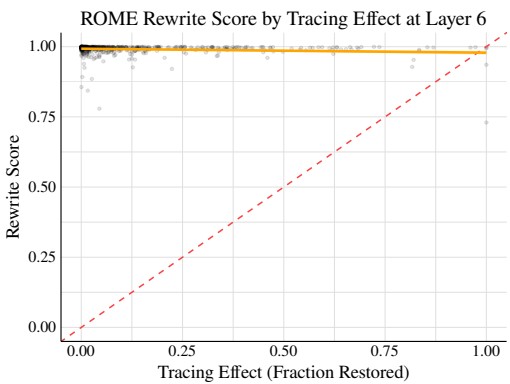

Figure 4: The correlation between ROME edit success and the tracing effect at layer 6 in GPT-J is not positive but in fact slightly negative ($\rho = -0.13$; $p < 1e-3$). The dashed red line shows a hypothetical perfect relationship.

subset of facts that are correctly completed by GPT-J, in order to ensure that there is knowledge to localize in the model for each point (details in Appendix A). Our final sample size is $n = 652$.

## 4.3 Experiment Results

We present results in two ways. First, in Fig. 4, we show Rewrite Score as a function of the (fractional) tracing effect. The red dotted line shows a hypothetical perfect relationship between tracing and edit success. Surprisingly, there is not a positive relationship but a *negative* relationship between the rewrite score and the tracing effect (linear correlation of $\rho = -0.13$; $p < 1e-3$). This seems to fully invalidate the assumption that editing should be most effective when it occurs at a layer where information is stored about the edited fact. We wish to emphasize, however, that in most layers we simply see a near-zero rather than negative correlation, as shown in Appendix Fig. 15.

Our second mode of analysis is though linear regression models predicting rewrite score based on (1) the tracing effect, (2) the choice of edit layer treated as a categorical variable, or (3) both terms interacted, again treating edit layer as a categorical variable. The purpose of the models is to show how much of the variance in rewrite score is explained by one variable versus the other. We show the resulting $R^2$ values in Table 1. We see that the choice of layer explains almost all of the variance in

Table 1: $R^2$ values for predicting ROME edit success. Tracing effects explain essentially none of the variance in rewrite score, while the choice of edit layer is very important.

|  | $R^2$ Values | | |
|---|---|---|---|
| Method | Layer | Tracing Effect | Both |
| ROME | 0.947 | 0.016 | 0.948 |

rewrite score (94.7%), while adding the tracing effect to the model raises the $R^2$ only to 94.8%. This means that **the tracing effect is able to explain only 0.1% of the variance in edit success** when accounting for the choice of edit layer. These results suggest that the tracing effect is essentially unrelated to the success of model editing.

This is a surprising conclusion, and it naturally raises the question of why applying ROME at layer 6 works well in the first place (see average rewrite, paraphrase, and neighborhood scores across layers in Appendix Fig. 7). We suggest a possible answer to this question in Sec. 6.

| Editing Problem Variants | Input Prompt | Objective |
|---|---|---|

Error Injection — Autonomous University of Madrid, which is located in ___________ ⟶ $\arg\max_{\theta} p_\theta(\text{Sweden}|\text{Input})$

Tracing Reversal — Autonomous University of Madrid, which is located in ___________ ⟶ $\arg\max_{\theta} p_\theta(o_{\text{noise}}|\text{Input})$

Fact Erasure — Autonomous University of Madrid, which is located in ___________ ⟶ $\arg\min_{\theta} p_\theta(\text{Spain}|\text{Input})$

Fact Amplification — Autonomous University of Madrid, which is located in ___________ ⟶ $\arg\max_{\theta} p_\theta(\text{Spain}|\text{Input})$

Fact Forcing — *Autonomous University of Madrid*, which is located in ___________ ⟶ $\arg\max_{\theta} p_\theta(\text{Spain}|\text{Noisy Input})$
        ‿‿‿‿‿‿‿‿‿‿‿‿‿‿‿‿‿‿‿‿
        Add noise to subject

Figure 5: Depiction of editing problem variants. Rather than inject a new false fact into a model (Error Injection), we consider injecting the output obtained from noising the subject entity (Tracing Reversal), erasing a stored fact (Fact Erasure), amplifying a stored fact (Fact Amplification), or forcing a known fact onto the same kind of noisy input as used in Causal Tracing (Fact Forcing).

**Additional Robustness Experiments**. We include additional results in Appendix B using another dataset, ZSRE [20] (Figs. 19 and 20, Table 8), and another localization method, representation zeroing [2] (Figs. 21 and 22). Further robustness experiments in Appendix C include results with (1) other measures of edit success including Paraphrase Score, Neighborhood Score, and an Overall Score (Tables 4, 5 and 6), (2) different values of the tracing window size (Fig. 12), (3) GPT2-XL rather than GPT-J (Fig. 13), (4) the original unscaled metrics from Meng et al. [21] (Fig. 14), and (5) tracing effects measured at the last subject token rather than the max across tokens (Fig. 16). We find that **all of these experiments corroborate our results comparing Causal Tracing to Rewrite Score for GPT-J on CounterFact.** Considering these robustness results alongside additional editing method experiments that we consider in Sec. 5 below, we note that our main conclusions generalize across different causal localization methods, editing methods, editing metrics, models, and datasets.

## 5 Reconciling Localization and Editing

If injecting a new fact has little to do with where an existing fact is stored in the model, perhaps there is some other editing intervention that would be more closely related to insights from tracing analysis. In this section, we propose a few variants of the model editing problem that appear more and more like Causal Tracing in terms of their input, target, and objective. Then, we repeat and extend our analysis from Sec. 4 for all of these editing problems.

### 5.1 Editing Problem Variants

We summarize the following editing problems in Fig. 5.

1. *Error Injection.* The editing problem considered in Sec. 4, the objective being to maximize $p_\theta(o_{false}|s, r)$.
2. *Tracing Reversal.* We maximize $p_\theta(o_{noise}|s, r)$, aiming to change the model output from $o_{true}$ back to the output for the "original" noised input $P = (s_{noise}, r)$ in Causal Tracing, $o_{noise}$.
3. *Fact Erasure.* Knowing where a fact is stored could be more useful for erasing the fact rather than injecting a new one. Hence, we consider erasing a fact by minimizing $p_\theta(o_{true}|s, r)$.
4. *Fact Amplification.* We reinforce known facts in the model by maximizing $p_\theta(o_{true}|s, r)$. Even for correctly predicted points, this value is often not near 1, leaving room for it to be increased.
5. *Fact Forcing.* As in Causal Tracing, this method uses a noised subject representation $s_{noise}$. We force the model to output $o_{true}$ for this input by maximizing $p_\theta(o_{true}|s_{noise}, r)$. Though this problem is of little practical significance, it is the most similar to Causal Tracing in its design, since it uses the same input as Causal Tracing and matches the goal of increasing the probability of $o_{true}$ (see Sec. 3.2).

Note that solutions to each of these problems are evaluated according to our Rewrite Score, Paraphrase Score, and Neighborhood Score metrics from Sec. 3.4. The only difference is in the target output for the rewrite and paraphrase metrics (neighborhood is entirely identical).

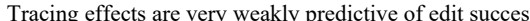

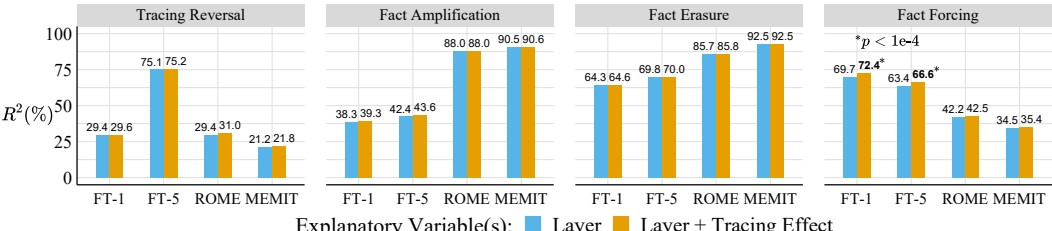

Figure 6: Tracing effects are very weakly predictive of edit success across editing problems and methods. Relative to the $R^2$ of a regression predicting rewrite score based on the edit layer (blue), a regression with edit layer and tracing effects (orange) improves the $R^2$ by at most .03 points (bolded). The choice of edit layer is a much better predictor of the rewrite score.

## 5.2 Experiment Design and Additional Edit Methods

We use the same experimental procedure as in Sec. 4, except that we consider a broader set of editing methods besides ROME. We list the four methods below:

1. ROME. The edit method from Sec. 4, ROME edits a single MLP layer's down-projection weight.
2. MEMIT. Though designed to edit multiple facts at once, when editing a single fact this method differs from ROME only by spreading out its update over several layers rather than one layer [22].
3. Constrained Finetuning (window size 1). We adopt a simple Adam-based optimization approach with an $\ell_\infty$-norm constraint, following Zhu et al. [42]. The window size of 1 indicates we apply this method at a single layer.
4. Constrained Finetuning (window size 5). The above finetuning method on five adjacent layers.

We select these methods for their simplicity and since ROME and MEMIT are designed specifically to edit MLP layers. Note that we report results for Causal Tracing with a window size of five, so when we use MEMIT or constrained finetuning to edit five layers, these five layers can exactly match the range of restored layers from Causal Tracing.

## 5.3 Experiment Results

**Main Results.** As in our analysis in Sec. 4, we report $R^2$ values for a linear regression model predicting the rewrite score based on (1) the choice of edit layer treated as a categorical variable, or (2) that variable interacted with the tracing effect. We show the results in Fig. 6, with $R^2$ values for each regression above their respective bars (numbers also in Appendix Table 3). We find that, relative to the Layer-only regression, **tracing effects explain at most an additional 3.2% of the variance in edit success** across our different editing problems and editing methods. This is a very small effect, especially compared to $R^2$ values from the Layer-only regression, which explains most of the variance in the outcome (58.5% on average across conditions in Fig. 6). We believe this is surprising given how the editing problem variants are designed. **It would seem that knowing where a fact is stored should help with amplifying or erasing that fact, but our results appear to fully disconfirm this hypothesis.** Interestingly, it also appears that it makes little difference whether we edit at one layer or five layers in order to match the number of representations restored by Causal Tracing. Based on comparisons between finetuning methods (FT-1 and FT-5) and between ROME and MEMIT (applied to 5 layers), editing at five layers does not improve the alignment between tracing and editing. In addition to our robustness results listed in Sec. 4.3, we also repeat our analysis using a subset of points where tracing effects are concentrated to a small number of layers, in order to focus on points where MEMIT and FT-5 edit *all* of the layers where the fact is stored. Results are nearly identical for this subset of the data (see Appendix B).

**One Successful Case.** We see the strongest positive relationship between edit success and tracing effects for Fact Forcing with finetuning methods. Here, we find that tracing effects explain an additional 3% of the variance in edit success (up from 1.5% for other experiments). This effect

is statistically significant at $p < 1\mathrm{e}{-4}$ according to an F-test[2] comparing the two models (see visualization in Appendix Fig. 17). The result for Fact Forcing suggests that using $s_{noise}$ rather than $s$ in the model input is the cause of the positive relationship between editing and localization. We rule out the choice of target and maximizing vs. minimizing the target probability as possible causes based on the design of each problem variant (see Fig. 5): (1) the choice of target is not important since results are similar for Error Injection, Tracing Reversal, and Fact Amplification, and (2) maximizing vs. minimizing the target probability is not important since results are similar for Fact Erasure and Fact Amplification. Yet, tracing effects are still weakly informative of Fact Forcing editing if they explain only 3% of the variance in edit success. This points to there being other deeper reasons for localization results being unrelated to editing success.

## 6 Discussion

**Does Causal Tracing tell us anything?** We show that Causal Tracing is not indicative of which layer to select for model editing. However, this does not mean that localization insights from Causal Tracing have been useless. Causal Tracing has helped reveal the role that early-to-mid-range MLP representations *at the last subject token index* play in factual association in autoregressive language models, and ROME does perform better on average when optimizing the last subject token representation rather than another token representation [21].[3] Past work finds that both MLP and attention layers can show large Causal Tracing effects, and additional empirical editing experiments then demonstrate that it is preferable to edit MLP weights [21].

**Why is edit success high at layers where the edited fact is not actually stored?** First, we note that information is gradually accumulated across layers in a Transformer forward pass, as discovered by past work [31, 12, 21, 22, 13]. We suggest that it is possible to "override" the information in layer $\ell$ with an edit to another layer $k$ (where $k < \ell$ or $k > \ell$). Since ROME is typically effective across a large range of layers (see Fig. 9), it appears that ROME can override the information accrued across 5 or 10 layers of a forward pass with an edit to a single layer outside of that range of layers. We summarize this hypothesis as follows: *Many layers could store a fact, and it happens that some do.*

If this hypothesis were true, it would be surprising because one cannot arbitrarily swap layers in a Transformer model without greatly damaging model performance [39]. That is, it should matter where information enters the residual stream, since later layers strongly depend on receiving the right incoming information from prior layers. We leave it to future work to further investigate this hypothesis.

**What do our results imply about using model editing to validate localization claims?** We interpret our results to suggest that Causal Tracing *answers a different question* than model editing does. That is, Causal Tracing answers a question about where factual information is carried in representations in a Transformer forward pass, and this question turns out to be a different question than the *editing* question of where is best to intervene in the Transformer in order to change the factual information it expresses. It seems critical, then, to carefully formalize the questions that one wishes to answer before (1) validating the results of localization via editing or (2) motivating the design of an editing method via localization, because the conclusions that can be drawn from a particular localization method might not be relevant for the performance of a given model editing method. This would not imply the conclusions from the localization analysis are invalid, though. For instance, we believe Causal Tracing reveals interesting insights about where MLP representations contain factual information (see Figs. 1 and 2). We only wish to suggest that localization analysis might answer a different question than the question answered by model editing.

These observations may have implications for the array of studies that validate their localization analysis by manipulating a certain model behavior via an intervention on the model component recommended by the analysis [29, 18, 2, 1, 26, 34, 8, 19, 7, 36, 4, 21]. Do model editing experiments provide *additional* evidence for claims about which model components are responsible for certain behaviors? If localization and editing answer different questions, editing experiments will not provide further evidence for localization conclusions.

---

[2]This tests if one model explains more of the variance than another model which has only a subset of the first's covariates (here, tracing effect and edit layer vs. only edit layer).

[3]Although, datapoint-level regression would provide stronger evidence that tracing effects predict which token representation is best to optimize with ROME (and rule out other confounders such as the edit layer).

## 7 Conclusion

We obtain the surprising result that model edit success is essentially unrelated to where factual information is stored in models, as measured by Causal Tracing. Faced with this result, we attempt to reconnect tracing-based localization with edit success by introducing four variants of the Error Injection problem using the CounterFact dataset. We find that edit success and tracing effects correlate best in our Fact Forcing setting. However, even in this case, tracing effects explain only a small fraction of the variance in editing performance, while the choice of edit layer is a much more important factor. This suggests that, counterintuitively, better mechanistic understanding of how pretrained language models work may not always translate to insights about how to best change their behavior.

## 8 Limitations

We note a few limitations of the experiments conducted in this paper:

(1) We work only with the CounterFact and ZSRE datasets, which we use as short English prompts with factual completions corresponding to a specific set of relations between subject and object entities. This is a basic form of factual knowledge, and localization and editing analysis may yield different trends for other forms of knowledge.

(2) We work with two autoregressive Transformers chosen for their representativeness of large language models that show a capacity for expressing factual knowledge in response to natural language prompts. However, the conclusions from our analysis may not generalize to models larger than GPT-J (6B parameters) that are known to exhibit phase changes in their behavior under prompting.

(3) We use a particular set of localization and editing methods, including representation denoising and zeroing at the layer level and layer-level MLP editing methods that inject new facts or amplify or erase existing facts. Our conclusions may not necessarily hold for the breadth of localization and editing methods from work related to this paper, and one should be cautious in applying our conclusions beyond our experimental setting.

## 9 Broader Impacts

It is possible that increased mechanistic understanding of models improves our ability to edit them at some point in the future. In fact, we consider it unlikely that interpretability results never give insight into improving model editing methods. Thus, to the extent that model editing is a dual use methodology, which could be used to inject harmful beliefs or dangerous knowledge into models, interpretability results may enhance the effectiveness of these malicious use cases. However, these concerns are relatively far removed from our analysis, which focuses on the connection between localization and editing performance. Ultimately, we hope that studies of mechanistic interpretability and model editing improve our ability to control language models.

## Acknowledgements

We thank Kevin Meng and David Bau for helpful discussions of the methods used in this paper and the conclusions we reach from our experiments. Additionally, we thank Jasmijn Bastings and Lucas Dixon for feedback on writing and presentation of results, as well as the paper's reviewers for their useful comments. This work was conducted while Peter Hase was a student researcher at Google.

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

# A  Experiment Details

**Data Licenses.**  CounterFact is available by the MIT license at https://github.com/kmeng01/rome [21], and ZSRE is available publicly at http://nlp.cs.washington.edu/zeroshot/ [20].

**Data Filtering.**  We filter the CounterFact dataset to a subset of facts that are correctly completed by GPT-J, in order to ensure that there is knowledge to localize in the model for each point. We mark a completion correct when $o_{true}$ appears among the first 36 tokens sampled from the model given the prompt $P$ using greedy decoding. GPT-J achieves a completion accuracy of 32.6% under this scheme, and after starting with about 10% of the CounterFact dataset, our final sample size is $n = 652$. We perform additional filtering specifically for model editing in the Fact Erasure condition, where we filter points to have a target probability $p_\theta(o_{true}|s, r)$ of at least .02, so that there is a reasonable amount of probability mass to be erased. In this condition, we have $n = 489$ points.

**Compute.**  Experiments were run on a single NVIDIA A6000 GPU with 48gb memory. Computing editing performance for $n = 652$ points with GPT-J for a single edit method applied across model layers in the set {1, 5, 9, 13, 17, 21, 24, 28} could take about eight hours. Saving causal tracing or representation zeroing results for these datapoints takes about twelve hours. Regression analyses and plots can be made on demand (code in supplement) given the data from the editing and localization experiments.

**Edit Method Tuning.**  We tune the edit methods to have high rewrite scores while not trading off too aggressively against paraphrase and neighborhood scores. More specifically, this means we tune methods to have rewrite scores no higher than 99% (note methods can easily get above 99% rewrite score), separately for each editing problem variant. The tuning is done with the first 100 points of the CounterFact dataset, editing layer 6 for GPT-J and 18 for GPT2-XL. For ROME and MEMIT methods, we tune over the KL regularization weight values in the set {.0625, .9, 1}. For constrained finetuning, we tune over the $L_\infty$ norm weight values in the set {1e-4, 5e-5, 2e-5, 1e-5}. For both methods, we adopt default parameters from Meng et al. [22] unless otherwise stated. We describe the relevant hyperparameters below, for GPT-J first:

1. *Error Injection*. FT-1: norm constraint of 1e-4. FT-5: norm constraint of 2e-5. ROME: regularization weight of 1. MEMIT: regularization weight of 0.9.
2. *Tracing Reversal*. FT-1: Norm constraint of 1e-5. FT-5: Norm constraint of 2e-5. FT-5: 2e-5. ROME: default parameters. MEMIT: default parameters.
3. *Fact Erasure*. FT-1: norm constraint of 1e-5. FT-5: norm constraint of 1e-5. ROME: default parameters. MEMIT: default parameters.
4. *Fact Amplification*. FT-1: norm constraint of 1e-5. FT-5: norm constraint of 1e-5. ROME: default parameters. MEMIT: default parameters.
5. *Fact Forcing*. Note that for all methods we decide to increase the number of gradient steps, as convergence takes longer for finetuning (from 25 to 50 steps) and for the gradient-based optimization for $v^*$ in ROME (from 20 to 25 steps). FT-1: norm constraint of 1e-4. FT-5: norm constraint of 1e-4. ROME: 25 gradient steps for finding $v^*$. MEMIT: default parameters (already set to 25 steps).

We run only the Error Injection and Fact Forcing conditions for GPT2-XL. Hyperparameters are as follows:

1. *Error Injection*. FT-1: norm constraint of 1e-3. FT-5: norm constraint of 1e-4. ROME: default parameters. MEMIT: default parameters.
2. *Fact Forcing*. FT-1: norm constraint of 5e-4. FT-5: norm constraint of 5e-5. ROME: default parameters. MEMIT: default parameters.

# B  Additional Results

**ZSRE Dataset.**  Here, we describe experiments with the ZSRE dataset, which is commonly used in past editing method papers [9, 24]. ZSRE includes naturalistic questions rather than prompts

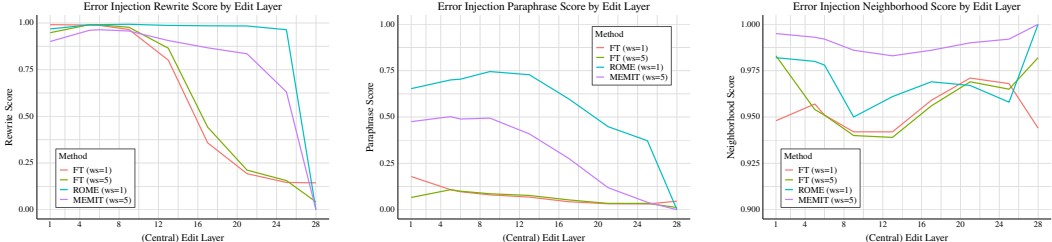

Figure 7: Edit success metrics for our four editing methods, under the Error Injection objective. Left: Rewrite, Center: Paraphrase, Right: Neighborhood.

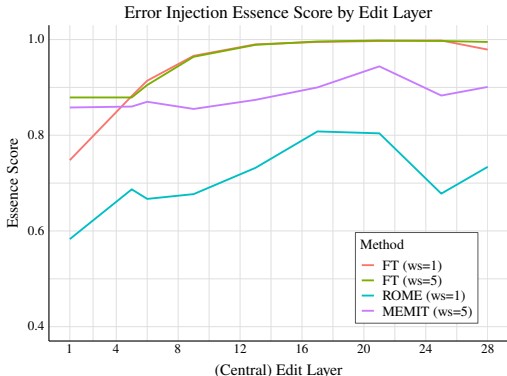

Figure 8: Essence score by edit layer, for our four editing methods, under the Error Injection objective.

intended for autoregressive cloze completion, as in CounterFact. Following past work [21], we use GPT-J to answer ZSRE questions in a zero-shot manner, and we edit the model with ROME. We report results for ZSRE via plots of edit success vs. tracing effect in Figs. 19 (rewrite score) and 20 (overall score), accompanied by regression analysis results in Table 8. We find that results with ZSRE match our conclusions with CounterFact, as the results are quite similar to plots and regressions with CounterFact data. Tracing effects are not predictive of edit success.

**Representation Zeroing.** Representation zeroing is a common localization technique where neural activations are manually set to zero during a model forward pass [18, 2]. We implement a form of representation zeroing that is exactly like Causal Tracing, except instead of denoising already-noised representations, we set clean representations to zero. Specifically, we simply run a normal forward

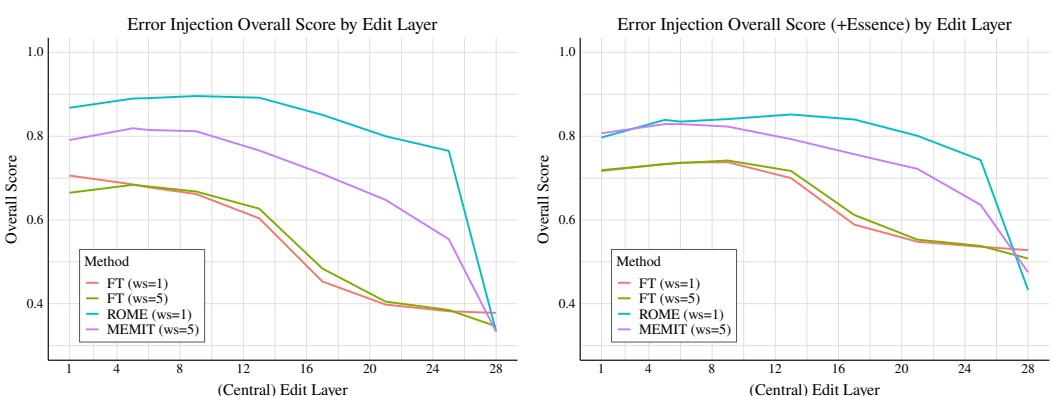

Figure 9: Overall edit success for our four editing methods, under the Error Injection objective. Left: The mean of Rewrite, Paraphrase, and Neighborhood Scores. Right: the mean score with Essence Score included.

Table 2: $R^2$ values for predicting ROME edit success in Error Injection, subsetted to 10% of the data that has the most concentrated tracing effects in a small number of layers. Even when facts appear to be stored at a small number of layers *and not other layers*, tracing effects are still not predictive of editing performance.

| Concentrated Data | $R^2$ Values | | |
| --- | --- | --- | --- |
| Method | Layer | Tracing Effect | Both |
| ROME | 0.927 | 0.02 | 0.929 |

pass until a certain set of layers (window size=5), where we zero out representation values for the MLP output representations at the subject token indices within those layers (then continue the forward pass). The localization effect is computed as the proportion of the original predicted probability that is deleted via the zeroing operation (ranging from no effect as 0% to 100% of probability deleted as 100%). These new results are shown in Figs. 21 for rewrite score and 22 for overall score, using ROME on GPT-J with CounterFact data. We obtain the same conclusions as our analysis with causal tracing: localization via representation zeroing is not predictive of edit success. Specifically, we see correlations between edit success and localization effect to be near zero across layers (using either rewrite score or overall score for edit success).

**Highly concentrated tracing effects.** Since Causal Tracing analysis suggests that information accrues gradually across layers (see Fig. 10), it seems possible that information is simply so diffusely spread across model layers that no matter what layer you edit, you will be editing a layer where a fact is at least stored in part. Based on this observation, we want to test whether tracing effects correlate better with edit success specifically when tracing effects are concentrated in a small number of layers. This condition represents that a fact appears to be stored in a small number of layers *and not elsewhere*. We hope that by editing in that range of layers, we can more easily manipulate that fact. To identify points with *concentrated* tracing effects, we use a heuristic for filtering points. Given the output of Causal Tracing analysis for a point, i.e. one effect per layer (the max across tokens), we define the point to have concentrated tracing effects when there are no more than three layers that have at least 50% of the maximum effect across layers (besides the layer with the max effect itself). Under this criterion, about 10% of the data (74 of 652 cases) have concentrated effects. Note we use our default tracing window size of 5 with the 28 layer GPT-J model for this experiment.

We show the results from our analysis on this data subset in Table 2, and **we observe no changes** in our main conclusions. For ROME with Error Injection, the added effect is 0.2%. Across editing problems and edit methods, the maximum added effect of including tracing effects on $R^2$ values for predicting rewrite score remains at 3.2% (for Fact Forcing with constrained finetuning). Thus, we conclude that even when facts appear to be stored in a small number of layers, localization results from Causal Tracing are still not informative about editing success, while the choice of edit layer is a far more important factor in whether a fact is successfully edited.

**Measuring essence drift.** Meng et al. [21] describe one possible consequence of model editing as *essence drift*, which occurs when core properties of an entity change after attempting to edit only one property of that entity. For example, changing where an island is located might also cause the model to nonsensically treat the island as a university campus (see example in Meng et al. [21]).

We aim to obtain an automatic metric to serve as a rough proxy for essence drift. A related metric is calculated with "Local Neutral" data involving the same subject entity but with other properties that are logically neutral with the original property of the subject being edited [15]. However, we do not have "Local Neutral" data for the CounterFact dataset, and essence drift aims to specifically measure changes to *core* properties of a subject.

Therefore, we automatically estimate changes to known properties of the subject $s$ by calculating the change in model perplexity over samples of text that were drawn from the pre-edit model given the prompt "$s$ is a " (which tend to describe a number of key properties of the subject $s$). We term these samples *essence texts*, and we obtain five samples per subject prompt by sampling with multinomial top-k sampling using $k = 5$. Given our essence texts, we measure the perplexity over the samples before and after editing a fact in the model, for every edited fact in our dataset. Note this is quite similar to the essence drift regularization objective used in the ROME optimization objective [21],

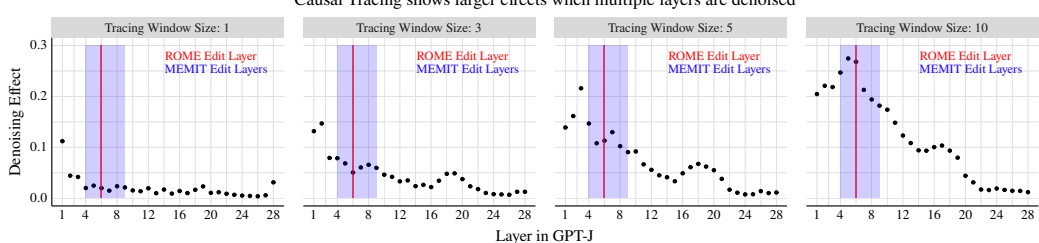

Figure 10: Tracing effects grow larger as the number of adjacent restored layer representations increases (tracing window size).

| Rewrite Score Table | | $R^2$ Values | | | | |
|---|---|---|---|---|---|---|
| Editing Problem | Method | Layer | Trace | Both | Diff | $p$-value |
| Error Injection | FT (1 layer) | 0.756 | 0.062 | 0.758 | 0.002 | <1e-4 |
| | FT (5 layers) | 0.775 | 0.055 | 0.777 | 0.002 | <1e-4 |
| | ROME (1 layer) | 0.947 | 0.016 | 0.948 | 0.001 | <1e-4 |
| | MEMIT (5 layers) | 0.677 | 0.024 | 0.678 | 0.001 | 0.199 |
| Tracing Reversal | FT (1 layer) | 0.067 | 0 | 0.067 | 0 | 0.997 |
| | FT (5 layers) | 0.751 | 0.045 | 0.752 | 0.001 | 0.032 |
| | ROME (1 layer) | 0.294 | 0.017 | 0.31 | 0.015 | <1e-4 |
| | MEMIT (5 layers) | 0.212 | 0.036 | 0.218 | 0.006 | <1e-4 |
| Fact Erasure | FT (1 layer) | 0.643 | 0.028 | 0.646 | 0.003 | <1e-4 |
| | FT (5 layers) | 0.698 | 0.025 | 0.70 | 0.002 | <1e-4 |
| | ROME (1 layer) | 0.857 | 0.019 | 0.858 | 0 | 0.555 |
| | MEMIT (5 layers) | 0.925 | 0.019 | 0.925 | 0 | 0.669 |
| Fact Amplification | FT (1 layer) | 0.383 | 0.014 | 0.393 | 0.01 | <1e-4 |
| | FT (5 layers) | 0.424 | 0.01 | 0.436 | 0.011 | <1e-4 |
| | ROME (1 layer) | 0.88 | 0.02 | 0.88 | 0 | 0.654 |
| | MEMIT (5 layers) | 0.905 | 0.018 | 0.906 | 0.001 | <1e-4 |
| Fact Forcing | FT (1 layer) | 0.697 | 0.104 | 0.724 | **0.027** | <1e-4 |
| | FT (5 layers) | 0.634 | 0.10 | 0.666 | **0.032** | <1e-4 |
| | ROME (1 layer) | 0.422 | 0.004 | 0.425 | 0.003 | <1e-4 |
| | MEMIT (5 layers) | 0.345 | 0.041 | 0.354 | 0.009 | <1e-4 |

Table 3: $R^2$ values for predicting **rewrite** score from choice of edit layer and tracing effect, across editing problem variants (corresponds to data in Fig. 6). Diff shows the added effect of including tracing in the regression (Both vs. Layer Only), in terms of $R^2$, and $p$-value shows the results from an F-test comparing the Both and Layer Only models. Tracing has some predictive value for Fact Forcing, but the $R^2$ value remains small compared to the choice of edit layer.

but we consider it as a metric here. We scale the change in perplexity to a fraction of 5, with the cut-off of 5 chosen to represent a maximally bad change to the model perplexity. Similar to our other metrics, our essence score is 1 if model perplexity on the essence texts does not change after editing the model (capping to 1 in cases of slight decreases in perplexity), and it is 0 if the perplexity increases by 5 or more.

We show essence scores for editing methods across layers in 8. Interestingly, the trend across layers for this metric is mostly counter to the trends for other metrics (Fig. 7), with editing later layers being generally preferable to editing earlier layers. As a result, when combined with the other metrics in Fig. 9, we see that the overall score trend flattens and shifts slightly toward mid-range layers in the model.

## C  Robustness Experiments

In addition to our main results with ROME for GPT-J and our Rewrite Score metric, we include robustness experiments to confirm that results are similar for (1) other measures of edit success including Paraphrase Score, Neighborhood Score, and Overall Score (Tables 4, 5, and 6), (2) different

Table 4: $R^2$ values for predicting **paraphrase** score from choice of edit layer and tracing effect, across editing problem variants. Diff shows the added effect of including tracing in the regression (Both vs. Layer Only), in terms of $R^2$, and $p$-value shows the results from an F-test comparing the Both and Layer Only models. The added effect of including tracing effects is very small across conditions (less than 3%).

| **Paraphrase Score Table** | | $R^2$ Values | | | | |
|---|---|---|---|---|---|---|
| Editing Problem | Method | Layer | Trace | Both | Diff | $p$-value |
| Error Injection | FT (1 layer) | 0.061 | 0.005 | 0.063 | 0.002 | 0.258 |
| | FT (5 layers) | 0.036 | 0.003 | 0.038 | 0.001 | 0.582 |
| | ROME (1 layer) | 0.279 | 0.001 | 0.303 | 0.024 | <1e-4 |
| | MEMIT (5 layers) | 0.246 | 0 | 0.269 | 0.023 | <1e-4 |
| Tracing Reversal | FT (1 layer) | 0.004 | 0.001 | 0.004 | 0 | 0.989 |
| | FT (5 layers) | 0.001 | 0 | 0.002 | 0.001 | 0.841 |
| | ROME (1 layer) | 0.01 | 0 | 0.012 | 0.002 | 0.121 |
| | MEMIT (5 layers) | 0.001 | 0 | 0.001 | 0 | 0.997 |
| Fact Erasure | FT (1 layer) | 0.046 | 0.001 | 0.048 | 0.002 | 0.303 |
| | FT (5 layers) | 0.079 | 0.007 | 0.084 | 0.005 | 0.004 |
| | ROME (1 layer) | 0.537 | 0.012 | 0.539 | 0.001 | 0.218 |
| | MEMIT (5 layers) | 0.586 | 0.015 | 0.587 | 0.001 | 0.184 |
| Fact Amplification | FT (1 layer) | 0.005 | 0.012 | 0.022 | 0.017 | <1e-4 |
| | FT (5 layers) | 0.017 | 0.013 | 0.035 | 0.018 | <1e-4 |
| | ROME (1 layer) | 0.24 | 0.002 | 0.267 | 0.027 | <1e-4 |
| | MEMIT (5 layers) | 0.236 | 0.001 | 0.263 | 0.026 | <1e-4 |
| Fact Forcing | FT (1 layer) | 0.044 | 0.004 | 0.046 | 0.002 | 0.367 |
| | FT (5 layers) | 0.023 | 0.002 | 0.025 | 0.002 | 0.387 |
| | ROME (1 layer) | 0.357 | 0.01 | 0.36 | 0.003 | 0.003 |
| | MEMIT (5 layers) | 0.095 | 0.001 | 0.105 | 0.01 | <1e-4 |

Table 5: $R^2$ values for predicting **neighborhood** score from choice of edit layer and tracing effect, across editing problem variants. Diff shows the added effect of including tracing in the regression (Both vs. Layer Only), in terms of $R^2$, and $p$-value shows the results from an F-test comparing the Both and Layer Only models. The added effect of including tracing effects is very small across conditions (2% or less).

| **Neighborhood Score Table** | | $R^2$ Values | | | | |
|---|---|---|---|---|---|---|
| Editing Problem | Method | Layer | Trace | Both | Diff | $p$-value |
| Error Injection | FT (1 layer) | 0.005 | 0 | 0.008 | 0.002 | 0.197 |
| | FT (5 layers) | 0.014 | 0.001 | 0.015 | 0.001 | 0.55 |
| | ROME (1 layer) | 0.011 | 0.003 | 0.015 | 0.005 | 0.001 |
| | MEMIT (5 layers) | 0.004 | 0.001 | 0.006 | 0.002 | 0.154 |
| Tracing Reversal | FT (1 layer) | 0.001 | 0 | 0.001 | 0 | 1 |
| | FT (5 layers) | 0.001 | 0 | 0.002 | 0.001 | 0.946 |
| | ROME (1 layer) | 0.001 | 0 | 0.002 | 0.001 | 0.946 |
| | MEMIT (5 layers) | 0.001 | 0 | 0.002 | 0 | 0.981 |
| Fact Erasure | FT (1 layer) | 0.01 | 0 | 0.014 | 0.004 | 0.037 |
| | FT (5 layers) | 0.01 | 0 | 0.013 | 0.004 | 0.06 |
| | ROME (1 layer) | 0.04 | 0.005 | 0.046 | 0.006 | 0.001 |
| | MEMIT (5 layers) | 0.05 | 0.007 | 0.059 | 0.009 | <1e-4 |
| Fact Amplification | FT (1 layer) | 0.012 | 0.009 | 0.02 | 0.008 | <1e-4 |
| | FT (5 layers) | 0.016 | 0.008 | 0.025 | 0.009 | <1e-4 |
| | ROME (1 layer) | 0.04 | 0.01 | 0.05 | 0.01 | <1e-4 |
| | MEMIT (5 layers) | 0.035 | 0.008 | 0.044 | 0.01 | <1e-4 |
| Fact Forcing | FT (1 layer) | 0.054 | 0 | 0.057 | 0.003 | 0.03 |
| | FT (5 layers) | 0.019 | 0.001 | 0.022 | 0.004 | 0.011 |
| | ROME (1 layer) | 0.299 | 0.022 | 0.311 | 0.012 | <1e-4 |
| | MEMIT (5 layers) | 0.046 | 0.012 | 0.066 | 0.02 | <1e-4 |

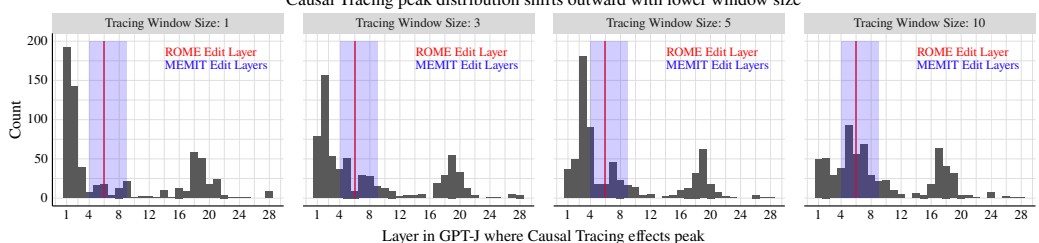

Figure 11: Each individual plot shows the distribution of tracing curve peaks (the argmax layer) across datapoints, using a different tracing window size. Together, the plots show how the distribution of layers where the tracing curves peak for each point shifts outward toward the first and last layer of the model as the tracing window size declines. This is primarily due to a clipping effect from using a window size greater than 1. The way tracing values are computed, a window size of 10 implies that the effect for "layer 1" is from restoring layers 1-5, while the effect for layer "layer 5" is 1-10. As a result, a tracing window size of 10 favors layer 5 over layers 1-4, and reducing the tracing window size leads to these clumps of effects shifting from layer 5 toward layer 1 (and from layer 24 to layer 28)

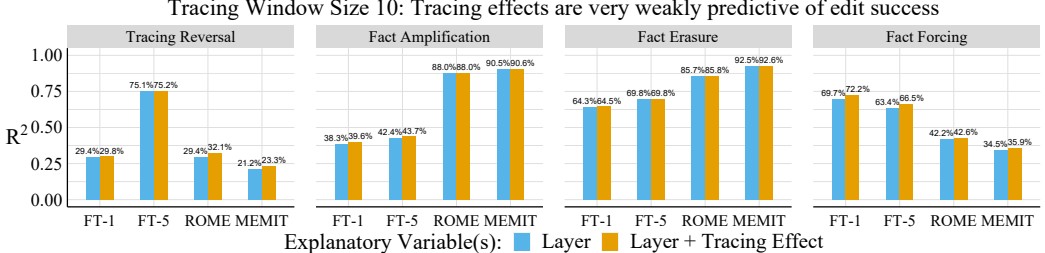

Figure 12: The results of our $R^2$ analysis for predicting rewrite score are nearly identical between using a tracing window size of 5 (shown in Fig. 6) or 10 (shown here).

values of the tracing window size (Fig. 12), (3) GPT2-XL rather than GPT-J (Fig. 13), (4) the original unscaled metrics from Meng et al. [21] (Fig. 14), and (5) using the tracing effect at the last subject token rather than the max across tokens (Fig. 16). We consider the last subject token effect since this corresponds more directly to the motivation for ROME (see Meng et al. [21]). We expand on each of these experiments below:

**Results for Paraphrase, Neighborhood, Overall Metrics.** We recreate our regression-based analysis across editing problem variants and editing methods using paraphrase score and neighborhood score as our outcomes rather than Rewrite Score, as well as an Overall Score that is the raw average of the three edit scores. These results are shown in Tables 4, 5, and 6 respectively. Similar to our analysis with rewrite score, these tables show that tracing effects are barely predictive of edit success at all. For paraphrase score, the largest gains in $R^2$ values are around 0.03 (relative to the layer-only regression model), and for neighborhood score, the largest gain is 0.02. The largest gain for overall score is 0.02 for Fact Forcing with constrained finetuning. Our overall conclusion remains that tracing effects are almost totally unrelated to edit success across editing problem variants, including for different edit success metrics.

**Results for Different Tracing Window Sizes.** We repeat our analysis from Sec. 5 using tracing effects obtained from a larger tracing window size of 10, to match the value used in Meng et al. [21]. Note that from Fig. 10, we know that the tracing effects grow larger as more adjacent layer representations are restored. When we recreate our main $R^2$ analysis using tracing effects with window size 10 (shown in Fig. 12), we find that results are nearly identical to those shown in Tables 3, 4, and 5.

**Results for GPT2-XL.** We rerun our analysis with GPT2-XL, a 48 layer model [30], while editing layers in the range {1, 5, 9, 13, 17, 18, 21, 25, 29, 33, 37, 41, 45, 48}. Here, we use a tracing window size of 10, and we limit our experiments to focus on Error Injection and Fact Forcing editing problems. As seen in Fig. 13, we find very similar trends when explaining rewrite score in terms of the choice

Table 6: $R^2$ values for predicting **overall** score (raw average of rewrite, paraphrase, and neighborhood scores) from choice of edit layer and tracing effect, across editing problem variants. Diff shows the added effect of including tracing in the regression (Both vs. Layer Only), in terms of $R^2$, and $p$-value shows the results from an F-test comparing the Both and Layer Only models. The added effect of including tracing effects is very small across conditions (2% or less).

| **Ovr. Edit Score** | | | $R^2$ Values | | | | |
|---|---|---|---|---|---|---|---|
| Editing Problem | Method | | Layer | Trace | Both | Diff | $p$-value |
| Error Injection | FT | (1 layer) | 0.642 | 0.054 | 0.643 | 0.002 | 0.001 |
| | FT | (5 layers) | 0.663 | 0.047 | 0.665 | 0.002 | 0.001 |
| | ROME | (1 layer) | 0.62 | 0.003 | 0.629 | 0.009 | <1e-4 |
| | MEMIT | (5 layers) | 0.525 | 0.008 | 0.534 | 0.009 | <1e-4 |
| Tracing Reversal | FT | (1 layer) | 0.294 | 0.025 | 0.296 | 0.002 | 0.054 |
| | FT | (5 layers) | 0.751 | 0.045 | 0.752 | 0.001 | 0.032 |
| | ROME | (1 layer) | 0.296 | 0.016 | 0.31 | 0.014 | <1e-4 |
| | MEMIT | (5 layers) | 0.21 | 0.036 | 0.216 | 0.006 | <1e-4 |
| Fact Erasure | FT | (1 layer) | 0.28 | 0.007 | 0.283 | 0.004 | 0.008 |
| | FT | (5 layers) | 0.119 | 0 | 0.124 | 0.004 | 0.015 |
| | ROME | (1 layer) | 0.718 | 0.023 | 0.718 | 0 | 0.729 |
| | MEMIT | (5 layers) | 0.794 | 0.025 | 0.794 | 0 | 0.555 |
| Fact Amplification | FT | (1 layer) | 0.188 | 0.003 | 0.199 | 0.011 | <1e-4 |
| | FT | (5 layers) | 0.224 | 0.002 | 0.236 | 0.013 | <1e-4 |
| | ROME | (1 layer) | 0.583 | 0.005 | 0.59 | 0.007 | <1e-4 |
| | MEMIT | (5 layers) | 0.597 | 0.005 | 0.607 | 0.01 | <1e-4 |
| Fact Forcing | FT | (1 layer) | 0.487 | 0.056 | 0.5 | 0.013 | <1e-4 |
| | FT | (5 layers) | 0.459 | 0.057 | 0.475 | 0.017 | <1e-4 |
| | ROME | (1 layer) | 0.285 | 0.004 | 0.291 | 0.006 | <1e-4 |
| | MEMIT | (5 layers) | 0.226 | 0.017 | 0.227 | 0.001 | 0.419 |

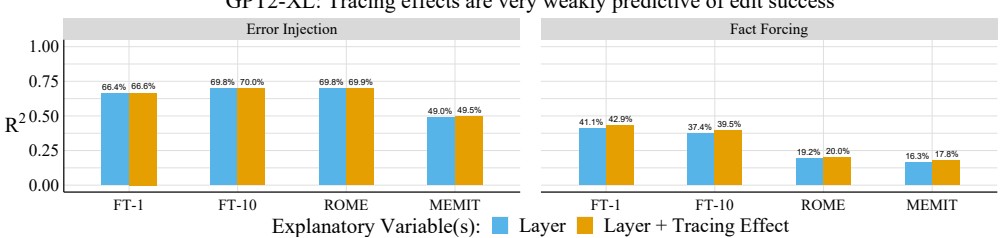

Figure 13: Like with GPT-J, tracing effects are very weakly predictive of edit success across editing problem variants for **GPT2-XL** while Fact Forcing shows the largest relationship. Relative to the $R^2$ of a model predicting rewrite score based on the choice of edit layer (blue), a model with edit layer and tracing effects (orange) improves the $R^2$ by at most .02 points for Fact Forcing. The choice of edit layer explains a much greater share of the variance in rewrite score.

of edit layer and the tracing effect at that layer. The largest explanatory effects in terms of $R^2$ are observed for Fact Forcing with constrained finetuning, but these effects remain small at about 2%.

**Results for Unscaled Metrics.** We repeat our analysis using the original editing metrics and absolute tracing effects from Meng et al. [21]. Their rewrite magnitude is the absolute difference between the probability of the new target $o_{false}$ and the old true target $o_{true}$ after editing, $p_{\theta^*}(o_{false}|s,r) - p_{\theta^*}(o_{true}|s,r)$. The tracing effect is the absolute tracing effect, $p_{\theta}(o_{true}|s_{noise},r,v_{(t,\ell)}) - p_{\theta}(o_{true}|s_{noise},r)$, measured at the last subject token index. We adjusted our rewrite and tracing metrics to (1) rely only on the target output probability, rather than difference in probabilities of two different targets which might not be appropriate for our different editing problems, and (2) to always fall between 0 and 1 for better comparability between datapoints, since absolute tracing effect are bounded by the original model probabilities. However, we reach the same conclusions from our analysis when using the original editing metrics. We show an example for rewrite magnitude and the

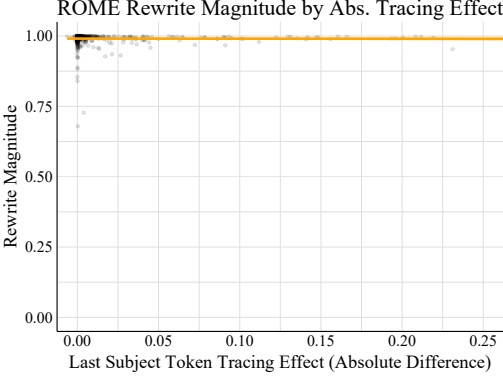

Figure 14: Editing vs. tracing results for ROME at layer 6 for Error Injection, using the un-rescaled rewrite and tracing metrics from Meng et al. [21]. Here, rewrite magnitude is the difference between the probability of the new target $o_{false}$ and the old true target $o_{true}$ after editing, $p_{\theta^*}(o_{false}|s,r) - p_{\theta^*}(o_{true}|s,r)$. The tracing effect is the absolute tracing effect, $p_\theta(o_{true}|s_{noise}, r, v_{(t,\ell)}) - p_\theta(o_{true}|s_{noise}, r)$, measured at the last subject token index. The correlation here is near zero, at $\rho = -.006$.

| Edit Metric | Regression Metric | Predictor(s) | Value |
|---|---|---|---|
| Rewrite Score | $R^2$ | Layer
Tracing Effect | 0.947
0.016 |
| | RMSE | Layer
Tracing Effect | 0.073
0.315 |
| | MAE | Layer
Tracing Effect | 0.02
0.206 |
| Overall Score | $R^2$ | Layer
Tracing Effect | 0.618
0.003 |
| | RMSE | Layer
Tracing Effect | 0.133
0.216 |
| | MAE | Layer
Tracing Effect | 0.11
0.183 |

Table 7: **Additional regression error metrics** (for CounterFact and ROME) lead us to the same conclusion as our analysis based on $R^2$. RMSE is root mean squared error, and MAE is mean absolute error. Regressions predicting rewrite score (or overall score) from the choice of edit layer achieve much lower prediction errors than regressions using the tracing effect, suggesting that the choice of edit layer is much more important for edit success than the tracing effect.

absolute tracing effect for Error Injection in Fig. 14. The correlation between edit success and tracing effect is still near zero.

**Results for Last Subject Token Effect.** ROME increases the target probability $p(o_{false}|s,r)$ by optimizing for a new output representation from a chosen MLP layer *at the last subject token index*. Meng et al. [21] show that this choice of token representation is critical to the success of the editing method, which is a hypothesis directly motivated by the results from their Causal Tracing analysis. In our paper, we by default report results using tracing effects that are the max across tokens at a given layer, for better comparability across the editing methods we use. However, when we repeat our analysis using the tracing effect specifically at the last subject token index, we obtain the same negative conclusions about the relationship between Causal Tracing localization and ROME editing performance. We show the correlations between Rewrite Score and Last Subject Token Tracing Effect in Fig. 16, where we see there are no positive correlations between editing success and tracing results at any layer in GPT-J.

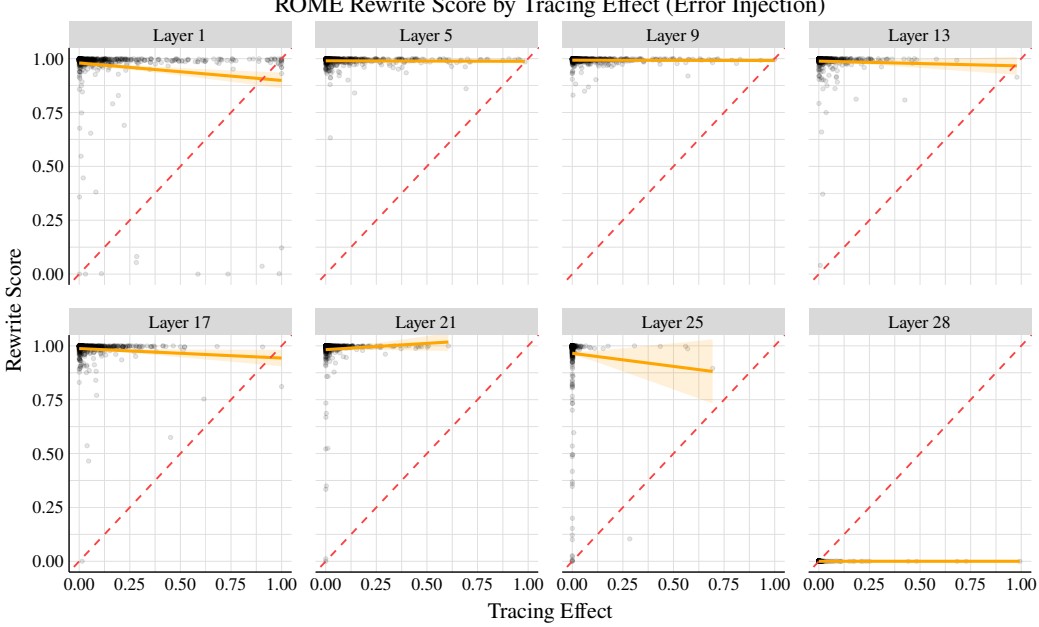

Figure 15: The relationship between ROME edit success and the tracing effect is near zero at most edit layers in the model (for the standard Error Injection editing problem). Red lines show perfect relationships between tracing effect and edit success.

| Edit Metric | Regression Metric | Predictor(s) | Value |
|---|---|---|---|
| Rewrite Score | $R^2$ | Layer
Tracing Effect | 0.795
0.042 |
| | RMSE | Layer
Tracing Effect | 0.158
0.341 |
| | MAE | Layer
Tracing Effect | 0.072
0.254 |
| Overall Score | $R^2$ | Layer
Tracing Effect | 0.654
0.059 |
| | RMSE | Layer
Tracing Effect | 0.136
0.223 |
| | MAE | Layer
Tracing Effect | 0.097
0.188 |

Table 8: **ZSRE regression results** lead us to the same conclusion as our experiments on CounterFact, using ROME editing. RMSE is root mean squared error, and MAE is mean absolute error. Regressions predicting rewrite score (or overall score) from the choice of edit layer achieve much lower prediction errors than regressions using the tracing effect, suggesting that the choice of edit layer is much more important for edit success than the tracing effect.

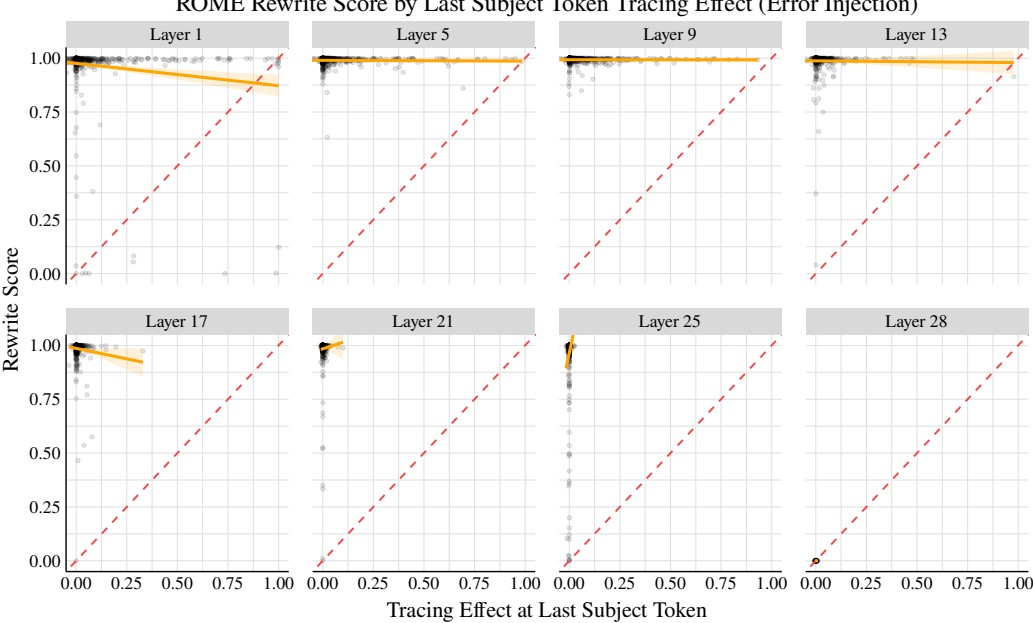

Figure 16: The relationship between ROME edit success and the tracing effect at the last subject token. The ROME method edits a fact by changing the output representation for the MLP layer specifically at the token index corresponding to the last subject token. However, editing performance and tracing effect at this position still do not positively correlate. Note the distribution of points along the $x$ axis changes depending on the choice of edit layer since the distribution of tracing effects is calculated from tracing effects *at that layer*.

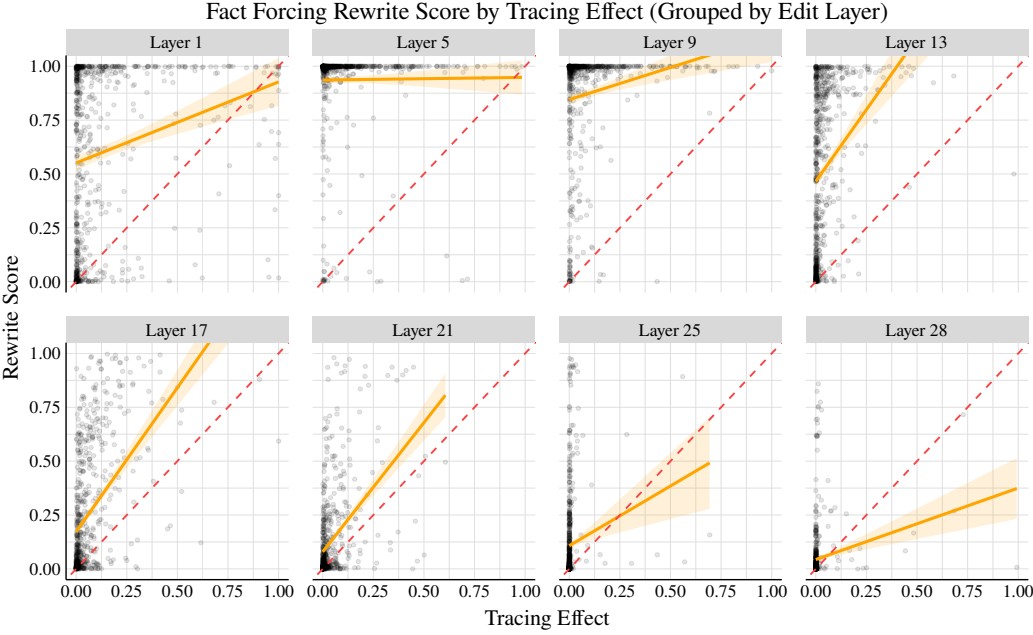

Figure 17: The relationship between Fact Forcing edit success and the tracing effect for constrained finetuning of 5 adjacent layers. "Layer $\ell$" indicates the center of this 5-layer interval, and the dashed red lines show a hypothetical perfect relationship between tracing effect and edit success. For many layers, there is a noticeable positive relationship between tracing effects and editing success. Yet, (1) there is a high amount of variance in the outcome, and (2) this variance is largely explained by the edit layer. As a result, tracing effects provide little extra information for predicting edit success beyond the choice of edit layer (about 3% more explained variance; see Fig. 6).

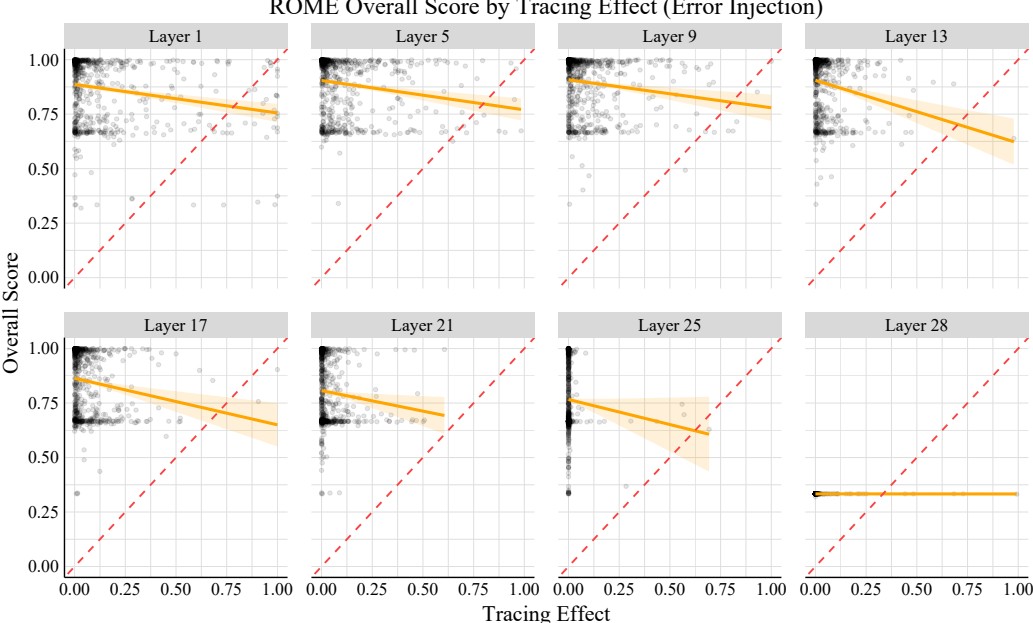

Figure 18: The relationship between ROME **overall score** (average of rewrite/paraphrase/neighborhood scores) and the tracing effect is somewhat negative for most edit layers in the model (for the standard Error Injection editing problem). Red lines show a perfect relationship between tracing effect and edit success, so a negative relationship suggests that tracing localization results do not indicate that editing will be successful.

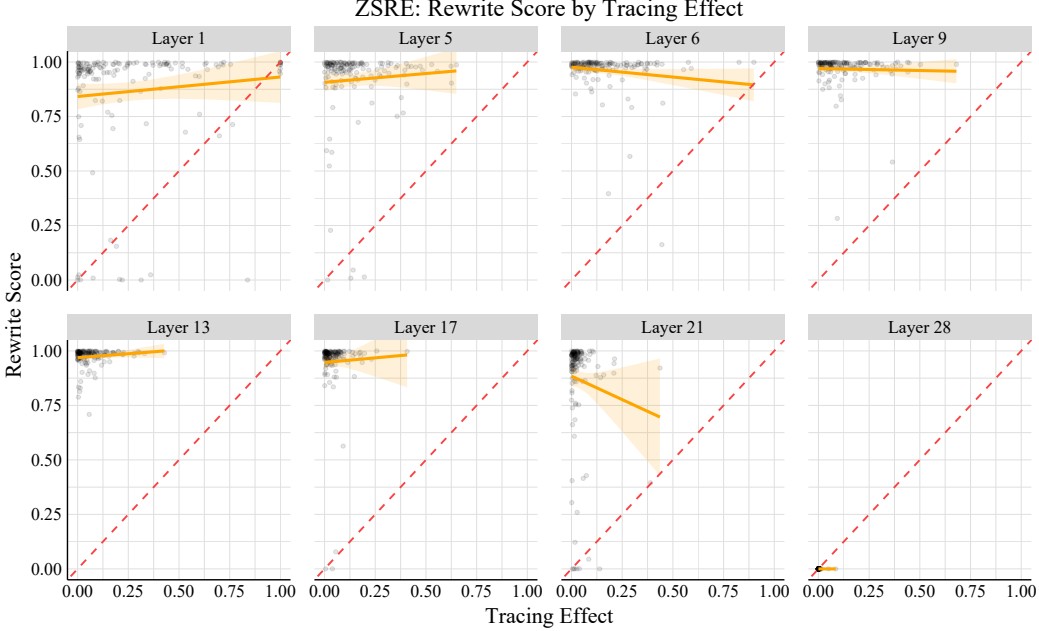

Figure 19: Additional experiments on the **ZSRE** dataset show the same results as for CounterFact, using the ROME editing method with rewrite score as our editing success metric (see regression analysis results in Table 8). Red lines show a perfect relationship between tracing effect and edit success, so near-zero relationships suggest that tracing localization results do not indicate that editing will be successful.

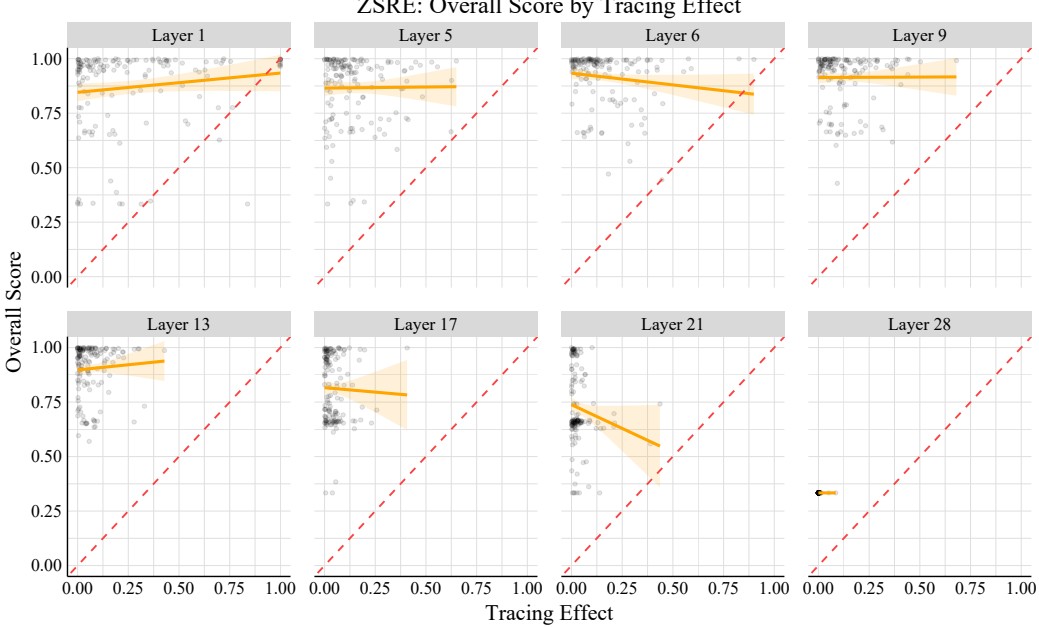

Figure 20: ZSRE experiments using overall score (average of rewrite/paraphrase/neighborhood scores) as the edit success metric.

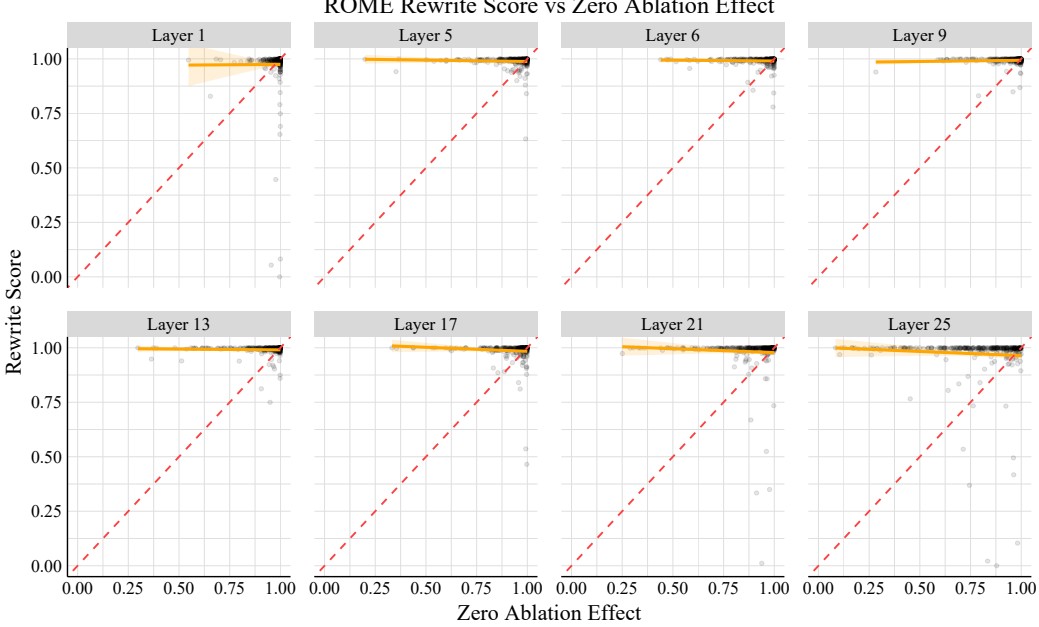

Figure 21: Additional experiments with **representation zeroing** as the localization method show the same results as for Causal Tracing, using the ROME editing method and **rewrite score** as the edit success metric. Red lines show a perfect relationship between representation zeroing and edit success, so near-zero relationships suggest that representation ablation localization results do not indicate that editing will be successful.

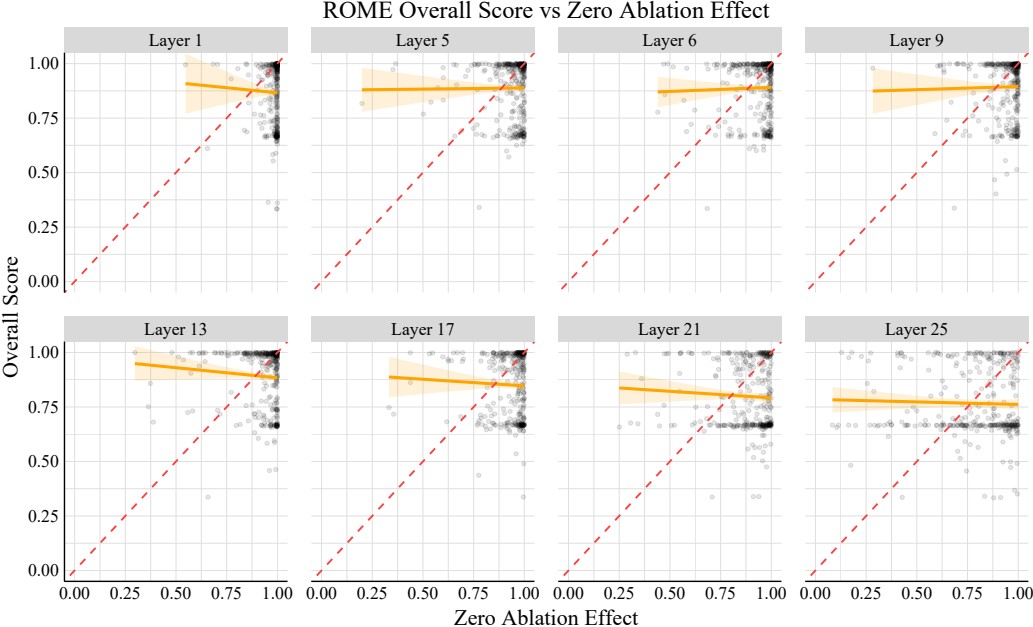

Figure 22: Additional experiments with **representation zeroing** as the localization method show the same results as for Causal Tracing, using the ROME editing method and **overall score** as the edit success metric. Red lines show a perfect relationship between representation zeroing and edit success, so near-zero relationships suggest that representation ablation localization results do not indicate that editing will be successful.

