# OpenReview forum: "Does Localization Inform Editing? Surprising Differences in Causality-Based Localization vs. Knowledge Editing in Language Models"
_NeurIPS.cc/2023/Conference — NeurIPS 2023 spotlight_

### Official Review · Reviewer_YdCx · 2023-06-30

**Soundness:** 4 excellent
**Presentation:** 4 excellent
**Contribution:** 4 excellent
**Rating:** 8
**Confidence:** 4

**Summary:**

The paper provides an experimental study on localization claims of causal tracing and editing methods such as ROME. Existing literature uses localization results to decide which layer to edit for knowledge editing in large language models. The paper shows that, surprisingly, localization and edit success are nearly uncorrelated. In fact, one can edit a large range of MLP layers to achieve editing success. Yet, localization suggests that only a few early-mid-layers store the factual information. The paper then proposes new tasks and datasets to reconcile this issue.

**Strengths:**

This is a very solid piece of empirical study. I find the conclusion convincing, as it is well supported by the experimental evidence.

The conceptual contribution is clear and novel. In particular, the uncorrelatedness of localization and edit success, as far as I know, has not been noticed by the literature. In fact, this is a somewhat surprising observation given the line of prior work such as ROME.

The paper is well-written.

**Weaknesses:**

I do not find major weakness of the paper. I have some minor comments and suggestions.

The Appendix contains additional experiments on zero ablation. Another technique one can try is mean ablation and resampling ablation, over a known datatset such as CounterFact. This may be more principled than zero ablation. See the [Causal Scrubbing](https://www.lesswrong.com/posts/JvZhhzycHu2Yd57RN/causal-scrubbing-a-method-for-rigorously-testing#4_Why_ablate_by_resampling_) work for some discussion. I believe this wouldn't make much difference here, but it's worth trying.

Figure 1: Is this produced by aggregating over the CounterFact datasets? Would it be possible to provide a fine-grained look into this? What are the facts that appear to be stored at later layers?

Line 145: Here, the noise level is chosen to be 0.094 universally. However, in ROME, this is chosen to be three times the observed standard deviation of embedding norms as sampled over a body of text. Note that this quantity depends on both the data and the specific (trained) model. Should we also follow the convention of ROME here, to make a fair comparison? In my experience, causal tracing is somewhat sensitive to the level of noise.  (I doubt it would change the conclusion, but it is worth looking into.)

Line 353: It might be worth a quick footnote here that not all examples have their tracing effect concentrated at the last subject token index. See Figure 11 of the original ROME paper for some natural instances.

Figure 10 in Appendix: y-axis should be Tracing Effect rather than Denoising Effect.

You used representation zero-ing in Appendix but representation zeroing in the main paper

**Questions:**

From a mechanistic interpretability point of view, could the author(s) provide more thoughts on the effectiveness of causal tracing (in locating which layer *mechanistically* stores the facts)?  In particular, would you argue that the model computation is really not as concentrated as causal tracing suggests? Is it rather a diffuse and incremental process, with lots of redundancy? It seems from the window size = 1 experiment (Figure 10 in Appendix), the picture clearly shows that no single layer gets much tracing effect. Thus, would you agree that the sliding window patching is not principled and could yield a false illusion that the information is sharply concentrated?

**Limitations:**

The author(s) addressed several limitations already. I believe it's adequate.

---

> ### Author Rebuttal · Authors · 2023-08-09
>
> Thanks for your review! We have a few comments below.
>
> > The Appendix contains additional experiments on zero ablation. Another technique one can try is mean ablation and resampling ablation, over a known dataset such as CounterFact. This may be more principled than zero ablation. See the Causal Scrubbing work for some discussion.
>
> Thanks for the suggestion. We agree this is an interesting kind of ablation for hidden states that would be worth trying in a localization method. Currently, we think a counterfactual-input ablation like in the Interpretability in the Wild paper (https://arxiv.org/pdf/2211.00593.pdf) would be ideal, because it seems like it should remove the task-relevant information in a representation while maintaining other information, like syntactic information. However, it seems like there is theoretical support for the resampling ablation approach (see Sec. 3 of https://arxiv.org/pdf/1910.13413.pdf), and it would be hard to obtain good “neutral” counterfactuals like in the Interpretability in the Wild paper in general (and in particular, we are not sure how to design such counterfactuals for factual association). Either way, we are a little concerned about the current noising ablation and zero ablation methods that have been popular to date, because these can produce OOD inputs to model layers (see a related discussion in Sec. 4 of https://arxiv.org/pdf/2106.00786.pdf). It will be interesting to see which ablation methods are most useful for model editing or circuit identification going forward.
>
> > Figure 1: Is this produced by aggregating over the CounterFact datasets? Would it be possible to provide a fine-grained look into this? What are the facts that appear to be stored at later layers?
>
> Yes this is from aggregating traces across our subsample of known facts in the CounterFact dataset (known to GPT-J). We did some qualitative analysis of our own but nothing made it into the paper because we couldn’t identify any noticeable trends. For some reason, many points have primarily late-layer MLP effects! This is one reason we think that localization methods remain interesting and potentially useful – they still reveal interesting trends about how models perform tasks.
>
> > Line 145: Here, the noise level is chosen to be 0.094 universally. However, in ROME, this is chosen to be three times the observed standard deviation of embedding norms as sampled over a body of text…Should we also follow the convention of ROME here, to make a fair comparison?
>
> The experiments should exactly match the settings from the ROME paper for GPT-J with CounterFact, where this .094 comes from, though it’s true that the results could be sensitive to this noise value with other models and datasets. We aren't immediately concerned given all the other robustness results we obtained. See our discussion of the Causal Scrubbing comment for the rest of our thoughts on noising ablations. (Basically, it seems potentially problematic that the noise is OOD to the hidden state distribution, although this is very much by design from choosing a three*sigma standard deviation.)
>
> > Line 353: It might be worth a quick footnote here that not all examples have their tracing effect concentrated at the last subject token index. See Figure 11 of the original ROME paper for some natural instances.
>
> True! We also noticed this, but it is worth emphasizing since it would be new information to some readers.
>
> > would you argue that the model computation is really not as concentrated as causal tracing suggests?…It seems from the window size = 1 experiment (Figure 10 in Appendix), the picture clearly shows that no single layer gets much tracing effect. Thus, would you agree that the sliding window patching is not principled and could yield a false illusion that the information is sharply concentrated?
>
> Yes we agree. That was one of the first things we thought when looking at how tracing effects changed with the window size. The window size 10 plots are a little misleading at first glance because of how concentrated they make the effects look, when in reality, it seems like information is gradually accrued across many layers.
>
> What’s interesting is that the model computation can be concentrated, in the sense that you can edit a single layer to achieve a new factual association. And it turns out if you run causal tracing again for a model after editing it, the MLP effects are much more concentrated. So there appears to be a meaningful difference between how concentrated the information is in a pretrained LM vs. an edited LM.

---

> > ### Comment · Reviewer_YdCx · 2023-08-10
> >
> > Thank you for the detailed response! I maintain my rating and recommend accept.
> >
> > For figure 1, I would suggest at least specify what is the subsample on which you produced the plot, and how many prompts there are. In practice, I find that small datasets may be affected by outliers. For example, certain prompts can be a bit resistant to Gaussian noise corruption of their embeddings. That is, the model still retains decent performance after Gaussian noise patching (with noise level chosen as in ROME).

---

### Official Review · Reviewer_FzCS · 2023-07-02

**Soundness:** 3 good
**Presentation:** 3 good
**Contribution:** 3 good
**Rating:** 7
**Confidence:** 3

**Summary:**

Focusing on recently popular model editing methods, the authors study whether localizing methods based on causal mediation analysis play a decisive role. Based on the sample-wise analysis of the tracing effect and rewriting score, they find some surprising results that model editing performances (rewriting score) are no correlation with the tracing effect and are more dependent on the choice of the layer. They also design other model editing performance metrics to verify their findings.

**Strengths:**

This paper talks about an important and interesting problem, the stability or robustness of the interpretability metric. This is why I scored it 6.

1. The metrics about localization (tracing effect) and model editing (rewriting score, generalization score, etc) are inconsistent: I had this question that why the authors choose different metrics for tracing effect and model editing performance when I read ROME. Recovering the correct object is different from modifying the original relation between the subject and object. The authors consider this point and also design more metrics to measure model editing performance from different perspectives.

2. The authors conduct comprehensive experiments to support their findings.

3. As mentioned in Summary part, these findings are surprising and will inspire further exploration of localizing and model editing methods.


**Weaknesses:**

1. The main conflict between this paper with ROME is the viewpoint to switching layers, which is also very important to understanding causal localizing for us:

    1.1 ROME says that "informed by the Zhao et al. (2021) finding that transformer layer order can be exchanged with minimal change in behavior, ......, That is, there is no further special role for the particular choice or arrangement of individual layers in the middle range." This is why ROME simply chooses one layer in early layers.

    1.2 This paper says that "If this hypothesis were true, it would be surprising because one cannot arbitrarily swap layers in a Transformer model without greatly damaging model performance [39]."

    1.3 However, the paper [39] just considers BERT model rather than GPT models and conducts experiments on MNLI dataset which is also different from the factual reasoning task. Therefore, I think the authors should conduct more analysis of GPT models and current task to support their viewpoints.

    1.4 I expect more analysis of why there are inconsistencies between causal localizing and rewriting after the authors help reveal them, or some guesses at least. That is why I only give 6.


**Questions:**

1. The first question is also 1.4  in the Weaknesses part.

2. As shown in Figure 5 in ROME, the authors also show model editing performance to support their causal tracing method. The metric used in y-axis is the portion cases for P(o^*) > P(o) (Page 7) rather than the sample-wise difference used by the authors. Does the author believe that inconsistency is a result of metric selection?

3. What is the author's opinion on current localization methods like the denoising method? When should one choose such methods? What inherent knowledge in the model did it exactly point out?

4. What is your opinion on the importance of the early layers and what leads it?

**Limitations:**

Please see Weakness part.

---

> ### Author Rebuttal · Authors · 2023-08-09
>
> Thanks for your review! We have a few comments below.
>
> > The main conflict between this paper with ROME is the viewpoint to switching layers…
>
> It’s true we disagree with the perspective in the ROME paper about how to interpret [39]. The paper [39] shows drops up to 6 percentage points of MNLI accuracy from switching adjacent layers in a BERT model. We think this is a big drop that suggests the order of layers in pretrained language models is important. As you note, the ROME paper says “there is no…special role for the particular choice or arrangement of individual layers in the middle range.” In our opinion, this isn’t really the right interpretation. We agree that both our paper and the ROME paper stretch the interpretation of [39] a bit, since it handles different models and tasks than factual association. However, this is a somewhat minor distinction in the end, since it’s just “motivation” for the ROME method and a discussion point in our paper. The ROME method works fairly well on the CounterFact benchmark, and we show that editing performance can be unrelated to localization results. We leave it to future work to further explore our conjecture in our Discussion setting: “Many layers could store a fact, and it happens that some do” (which we believe would be surprising in light of [39]).
>
> > I expect more analysis of why there are inconsistencies between causal localizing and rewriting after the authors help reveal them, or some guesses at least
>
> To be clear, this was our goal with Sec. 5 (analysis) and Sec. 6 (discussion).
>
> > As shown in Figure 5 in ROME, the authors also show model editing performance to support their causal tracing method. The metric used in y-axis is the portion cases for P(o^*) > P(o) (Page 7) rather than the sample-wise difference used by the authors. Does the author believe that inconsistency is a result of metric selection?
>
> We don’t think the results are sensitive to the precise choice of metric because we replicated our main result with the original Efficacy Magnitude metric from the ROME paper (see ln282 and Fig 14) as well as other editing metrics for paraphrase generalization and neighborhood specificity (see lns280-281, Tables 4, 5, and 6).
>
> > What is the author's opinion on current localization methods like the denoising method? When should one choose such methods? What inherent knowledge in the model did it exactly point out?
>
> We think that denoising and zeroing localization methods are interesting and potentially useful – they still reveal interesting trends about how models perform tasks (like in Fig. 1 and 2), specifically factual association in our context. That said, we currently think that a counterfactual-input ablation like in the Interpretability in the Wild paper (https://arxiv.org/pdf/2211.00593.pdf) would be better, because it seems like it should remove the task-relevant information in a representation while maintaining other information, like syntactic information. However, other approaches are available, like resampling replacement (see Sec. 3 of https://arxiv.org/pdf/1910.13413.pdf). It will be interesting to see which ablation methods are most useful for model editing or circuit identification going forward.
>
> > What is your opinion on the importance of the early layers and what leads it?
>
> First, we think the importance of early layers is slightly overstated due to results like our Figs. 1 and 10. We see that a substantial fraction of facts have their peak MLP effects in late layers, and more generally, the contribution of any single layer is quite limited. That said, there is good evidence that information accrues over time in the residual stream of Transformers (see paper citation: Transformer feed-forward layers are key-value memories), and in this view, it makes sense that it would be important for early layers to set the residual stream on the right track, such that later layers receive the necessary input and add the necessary information to the residual stream.

---

> > ### Comment · Reviewer_FzCS · 2023-08-11
> > **Thanks for your response.**
> >
> > I have read the author's response and am particularly satisfied with the answers to the last two questions.
> >
> > I also suggest that the authors should consider talking about The 1st question ("The main conflict between this paper with ROME is the viewpoint to switching layers…") in the revised version.
> >
> > I have raised my score to 7.
> >
> > The reviewer.

---

### Official Review · Reviewer_UfPL · 2023-07-06

**Soundness:** 4 excellent
**Presentation:** 4 excellent
**Contribution:** 4 excellent
**Rating:** 8
**Confidence:** 5

**Summary:**

This paper investigates whether causal localization, which "locates" factual associations within LLMs, provides locations in models which when modified have the greatest effect on those particular factual associations. Counter to intuitions and previous claims in the literature. the authors find that causality-based localization is **much** less predictive (~3 percent vs ~95 percent) than layer-choice for predicting the efficacy of editing factual associations (at MLPs at/across layers). They confirm this behaviour across model classes, localization techniques, layer-depth-windows, datasets and models. This finding is of significant import to the mechanistic interpretability community, as it lends credence to the possibility that models can be pushed toward certain associations in-spite of the presence of existing associations _for those same prompts_ which causal tracing would have identified. Indeed, in some cases using ROME (and other approaches) to inject associations at locations identified as relevant by causal tracing is _less performant_ than injecting them elsewhere.

**Strengths:**

This paper carefully scrutinizes a strongly held belief from previous works about "localization" and the storage of factual information in LLMs. Through precise problem formulation and many experimental validations, the authors are able to provide compelling evidence that previous assumptions were incorrect, and that choosing where to inject (/modify) association, versus finding where a model places assocations for use in a forward pass, are _meaningfully distinct_ questions.

Presentation, methodology and supplemental results are well-presented and placed within a coherent narrative. Any questions that arose throughout reading were addressed in the appendices or final discussion.

**Weaknesses:**

No major or minor weaknesses were identified.

## Nitpicks
* Lines 222 and 237 are duplicated
* Line 206 does not appear to relate to the _neighborhood score_, which measures the change in prediction for similar subjects; rather, it seems to be referring to a metric in which $s=s'$ with $s'\cup s* = \emptyset$

**Questions:**

None

**Limitations:**

The primary limitations would be concerns about generality of these findings. The authors carry out extensive experimental investigations along all likely-to-be-relevant choices of configuration, and address outstanding concerns about the relevance of such findings to interpretability more widely.

---

> ### Author Rebuttal · Authors · 2023-08-09
>
> Thanks for your review!
>
> > Line 206 does not appear to relate to the neighborhood score, which measures the change in prediction for similar subjects; rather, it seems to be referring to a metric in which…
>
> We meant for this line to describe the purpose of the neighborhood metric from the ROME paper, and we use the same “neighborhood” prompts from the CounterFact dataset to compute it. The s* is a subject that is similar to the original fact subject s only in the sense that they both have relation r to object o (from the original fact (s,r,o)). They are not necessarily similar in any sense besides sharing this property. See page 30 of https://arxiv.org/pdf/2202.05262.pdf for an example. But let us know if we misunderstood what you meant.

---

> > ### Comment · Reviewer_UfPL · 2023-08-14
> >
> > Dear Authors,
> >
> > Thank you for the additional clarification!
> >
> >  I retain the previous recommendation for strong acceptance of this paper.

---

### Official Review · Reviewer_q6TL · 2023-07-08

**Soundness:** 3 good
**Presentation:** 4 excellent
**Contribution:** 3 good
**Rating:** 7
**Confidence:** 4

**Summary:**

This paper investigates the assumption that we can edit the located component to modify the facutal knowledge stored in the language model. The assumption might be false due to the empirical findings of this paper: (1) many knowledge facts stored outside the located parts of the model, and  (2) the near-zero correlation between causal tracing edition and edit success.
This paper further proposes a variant Fact Forcing to improve the correlation between location and editing, while finds that editting the early and middle layers still performs better.

**Strengths:**

1.	This paper challenges the conventional belief that knowledge localization and editing are strongly correlated. It provides a new and interesting perspective that suggests the relationship between knowledge localization and editing may not be as strong as previously thought.
2.	The paper provides a comprehensive overview of the locate-then-edit approach for knowledge editing.
3.	The experiments conducted are detailed and thorough. The paper explores different localization methods, editing targets, and variations of editing methods. This comprehensive analysis provides valuable insights into the effectiveness and limitations of various approaches.


**Weaknesses:**

1.	The paper mentions a difference between the metrics used and Rome but lacks a specific explanation. Can you explain why your metrics differ from Rome?
2.	The paper discusses causal tracing methods in relation to fact forcing for editing goals but lacks tracing methods related to other editing goals. Can you consider incorporating tracing methods related to editing problem variants?


**Questions:**

See 1 and 2 in weakness.

3.	How do you view the storage of factual knowledge in GPT? According to the paper, should it be stored among a series of interconnected neurons? How should we design editing methods to address this characteristic? In other words, what insights does your conclusion provide for editing methods?

---

> ### Author Rebuttal · Authors · 2023-08-09
>
> Thanks for your review! We have a few comments below.
>
> >  Can you explain why your metrics differ from Rome?
>
> Certainly. (Note: we also check that our results hold with original ROME metrics, see Line282 and Fig 14.)
>
> So we make two changes from the ROME metrics, which are both designed to make the metrics more comparable across datapoints (mentioned briefly in Lines193-4). First, we drop the dependence on the “true label” from the original fact tuple (s,r,o). This is because ROME metrics like Efficacy Magnitude are not really comparable across datapoints, since for one datapoint the metric might take values in [-.5,.5] (when the post-edit p(o) is .5), while for another datapoint EM might take a value in [0, 1] (when the post-edit p(o) is 0). The fact that the metric takes a different range for different data makes it hard to reasonably aggregate scores across data.
>
> Now we could just use the post-edit p(o^c) as our metric (for counterfactual output o^c). But this runs into the issue that, if the pre-edit p(o^c) is 1, and our post-edit p(o^c) is .99, we’d be giving an editing method a high score for lowering the probability of the new target. Hence, we subtract off the pre-edit probability of p(o^c), then rescale to the [0,1] range to ensure comparability across datapoints (our final Rewrite Score). Altogether, we give an edit a score of 0 if it has no effect on the target probability, and we give it a score of 1 if it has the maximal desired effect on the target probability.
>
> > Can you consider incorporating tracing methods related to editing problem variants?
>
> It’s true that, besides Causal Tracing and Representation Zeroing, there could be tracing methods that look more like Fact Editing rather than Fact Forcing. In fact, one might take some inspiration from other interpretability methods that explain why a model outputs Y rather than Y’ for a given input, in terms of feature importance scores, like “Interpreting Language Models with Contrastive Explanations” (https://arxiv.org/pdf/2202.10419.pdf). This would be an interesting direction for future work on new localization methods, though we ultimately don’t think it would improve ROME editing scores. As a part of our view, we really think that the focus on new localization vs new editing methods will depend on the goal of the research: do you want to know how a model currently carries out some function, or how to change the model to perform some new function?

---

> > ### Comment · Reviewer_q6TL · 2023-08-16
> >
> > Thanks for your response. I retain my original score to recommend accepting this paper.

---

### Author Rebuttal · Authors · 2023-08-09

We would like to thank all of the reviewers for their detailed reading of the paper and thorough feedback. We are so glad to see such a positive reception for the work! Hopefully this paper can help foster much more research in the community about localization and model editing, as well as add nuance to the community’s understanding of editing methods such as ROME and MEMIT, which have been very popular. We look forward to answering any questions about the paper here and sharing in some discussion of possible future directions for research.

---

### Decision · Program_Chairs · 2023-09-21

**Decision:**

Accept (spotlight)

**Comment:**

This paper presents a commentary on ROME and related models that use localization for knowledge editing in Transformers.  Its experiments focus on whether edit success aligns with Causal Tracing. Evidence in this paper suggests this is not the case (Figure 4), even if we broaden the definition to include other formulations of editing (Figure 6).

The reviewers uniformly found the paper to be strong, tackling an interesting question with good experimental design and clear writing. I believe the paper is quite likely to influence researchers working in this area and make them re-examine the assumptions behind editing methods based on causal tracing.

The reviewers largely raised minor points or clarification questions, with the most major framing issue raised being about the interpretation of layer switching. However all of the issues were addressed in the discussion period
or are fixable in a camera-ready version.